# Learning in Non-Cooperative Configurable Markov Decision Processes

**Giorgia Ramponi**\*
ETH AI Center
Zurich, Switzerland
`gramponi@ethz.ch`

**Alberto Maria Metelli**
Politecnico di Milano
Milan, Italy
`albertomaria.metelli@polimi.it`

**Alessandro Concetti**
Politecnico di Milano
Milan, Italy
`alessandro.concetti@mail.polimi.it`

**Marcello Restelli**
Politecnico di Milano
Milan, Italy
`marcello.restelli@polimi.it`

## Abstract

The Configurable Markov Decision Process framework includes two entities: a Reinforcement Learning agent and a configurator that can modify some environmental parameters to improve the agent's performance. This presupposes that the two actors have identical reward functions. What if the configurator does not have the same intentions as the agent? This paper introduces the Non-Cooperative Configurable Markov Decision Process, a framework that allows modeling two (possibly different) reward functions for the configurator and the agent. Then, we consider an online learning problem, where the configurator has to find the best among a finite set of possible configurations. We propose two learning algorithms to minimize the configurator's expected regret, which exploit the problem's structure, depending on the agent's feedback. While a naïve application of the UCB algorithm yields a regret that grows indefinitely over time, we show that our approach suffers only bounded regret. Furthermore, we empirically validate the performance of our algorithm in simulated domains.

## 1 Introduction

The standard Reinforcement Learning [RL, 40] framework involves an agent whose objective is to maximize the reward collected during its interaction with the environment. However, there exist real-world scenarios in which the agent itself or an external supervisor (configurator) can *partially* modify the environment. In a car racing problem, for example, it is possible to modify the car setup to better suit the driver's needs. Recently, the Configurable Markov Decision Processes [Conf-MDPs, 29] were introduced to model these scenarios and exploit the configuration opportunities. Solving a Conf-MDP consists of simultaneously optimizing a set of environmental parameters and the agent's policy to reach the maximum expected return. In many scenarios, however, the configurator does not know the agent's reward, and their intentions are different, leading to new forms of interaction between the two actors. For instance, imagine we are the owner of a supermarket, and we have to arrange the products on the shelves. Our objective is to increase the company's final profit; on the other hand, a customer aims to spend the shortest time possible inside the supermarket and buy the indispensable products only. Since we do not know the customer reward function, the only possibility is to try different dispositions and observe the customers' reactions. What if we knew what buyers

---

\*Work done when Giorgia Ramponi was at Politecnico di Milano.

35th Conference on Neural Information Processing Systems (NeurIPS 2021).

are most interested in? In this case, we can *strategically* decide how to position other products close to the popular ones to induce the customer in a more profitable behavior for the supermarket owner.

In this paper, we model these scenarios introducing the Non-Cooperative Markov Decision Processes (NConf-MDP). This novel framework handles the possibility of having different reward functions for the agent and the configurator. While Conf-MDP assumes that the configurator acts to help the agent to optimize its expected reward, an NConf-MDP, instead, allows modeling a wider set of situations, including the cases in which agent and configurator display a non-cooperative behavior. Obviously, this setting cannot be addressed with straightforward application of the algorithms designed for *cooperative* Conf-MDP. In fact, if the configurator and the agent optimize separately different objectives, they might not converge to an equilibrium strategy [52, 12, 51, 13]. In this novel setting, we consider an online learning problem, where the configurator has to select a configuration, within a finite set of possible configurations, in order to maximize its own return. This framework can be seen as a *leader-follower* game, in which the *follower* (the agent) is selfish and optimizes its own reward function, and the *leader* (the configurator) has to decide the best configuration, based on its reward. Clearly, to adapt its decisions, the configurator has to receive some form of feedback related to the agent's behavior. We analyze two settings based on whether the configurator observes just the agent's actions or, in addition, a noisy version of the agent's reward.

**Contributions**   In this paper, we extend the Configurable Markov Decision Process setting to deal with situations where the configurator and the agent have different reward functions. We call this new framework the Non-Cooperative Markov Decision Process (NConf-MDP, Section 3). Then, we formalize the problem of finding the best environment configuration, according to the configurator's reward, as a leader-follower game, in which the agent (follower) reacts to each presented configuration with its best response policy (Section 4). We provide a first algorithm, Action-feedback Optimistic Configuration Learning (AfOCL), to tackle this problem under the assumption that the configurator observes the agent's actions only (Section 5.1). We show AfOCL achieves finite expected regret, scaling linearly with the number of admissible configurations. As far as we know, this represents the *first problem-dependent* regret analysis in a Multi-Agent RL setting. Then, we introduce a second algorithm, Reward-feedback Optimistic Configuration Learning (RfOCL), that assumes the availability of a noisy version of the agent's reward, in addition to the agent's actions (Section 5.2). We prove that, under suitable conditions, RfOCL further exploits the *structure* underlying the decision process, removing the dependence on the number of configurations. The analysis use novel ideas, combining the *suboptimality gaps* of the configurator with those of the agent. Finally, we provide an experimental evaluation on benchmark domains, inspired by scenarios that motivate the NConf-MDPs framework (Section 7). The proofs of the results presented in the paper are reported in Appendix B. A preliminary version of this work was presented at "AAAI-21 Workshop on Reinforcement Learning in Games" [36].

## 2   Preliminaries

A *finite-horizon Markov Decision Process* [MDP, 35] is a tuple $\mathcal{M} = (\mathcal{S}, \mathcal{A}, p, \mu, r, H)$ where $\mathcal{S}$ is a finite state space ($S = |\mathcal{S}|$), $\mathcal{A}$ is a finite action space ($A = |\mathcal{A}|$), $p : \mathcal{S} \times \mathcal{A} \times \mathcal{S} \rightarrow [0, 1]$ is the transition model, which defines the density $p(s'|s, a)$ of state $s' \in \mathcal{S}$ when taking action $a \in \mathcal{A}$ in state $s \in \mathcal{S}$, $\mu : \mathcal{S} \rightarrow [0, 1]$ is the initial state distribution, $r : \mathcal{S} \rightarrow [0, 1]$ is the reward function, and $H \in \mathbb{N}_{\geq 1}$ is the horizon. A stochastic decision rule $\pi_h : \mathcal{S} \times \mathcal{A} \rightarrow [0, 1]$ with $h \in [H]$ prescribes the probability $\pi_h(a|s)$ of playing action $a \in \mathcal{A}$ in state $s \in \mathcal{S}$. A stochastic policy $\pi = (\pi_1, \cdots, \pi_H) \in \Pi^H$ is a sequence of decision rules, where $\Pi^H$ is the set of stochastic policies over the horizon $H$.

A *finite-horizon Configurable Markov Decision* Process [Conf-MDP, 29] is defined as $\mathcal{CM} = (\mathcal{S}, \mathcal{A}, \mathcal{P}, \mu, r, H)$ and extends the MDP considering a configuration space $\mathcal{P}$ instead a single transition model $p$. The Q-value of a policy $\pi \in \Pi^H$ and configuration $p \in \mathcal{P}$ is the expected sum of the rewards starting from $(s, a) \in \mathcal{S} \times \mathcal{A}$ at step $h \in [H]$:

$$Q_h^{\pi, p}(s, a) = r(s) + \mathbb{E}_{s_{h'} \sim p, \pi} \left[ \sum_{h'=h+1}^{H} r(s_{h'})|s_h = s, a_h = a \right],$$

denoting with $\mathbb{E}_{s_{h'} \sim p, \pi}$ the expectation w.r.t. the state distribution induced by $\pi$ and $p$ at step $h'$. The value function is given by $V_h^{\pi, p}(s) = \mathbb{E}_{a \sim \pi_h(\cdot|s)}[Q_h^{\pi, p}(s, a)]$ and the expected return

is defined as $V^{\pi,p} = \mathbb{E}_{s \sim \mu}[V_1^{\pi,p}(s)]$. In a Conf-MDP the goal consists in finding a policy $\pi^*$ together with an environment configuration $p^*$ so as to maximize the expected return, i.e., $(\pi^*, p^*) \in \arg\max_{\pi \in \Pi^H, p \in \mathcal{P}} V^{\pi,p}$.

# 3 Non-Cooperative Conf-MDPs

The definition of Conf-MDP allows modeling scenarios in which agent and configurator share the same objective, encoded in a single reward function $r$. In this section, we introduce an extension of this framework to account for the presence of a configurator having interests that might differ from those of the agent.

**Definition 3.1.** *A Non-Cooperative Configurable Markov Decision Process (NConf-MDP) is defined by a tuple $\mathcal{NCM} = (\mathcal{S}, \mathcal{A}, \mathcal{P}, \mu, r_c, r_o, H)$, where $(\mathcal{S}, \mathcal{A}, \mathcal{P}, \mu, H)$ is a Conf-MDP without reward and $r_c, r_o : \mathcal{S} \to [0,1]$ are the configurator and agent (opponent) reward functions, respectively.*

Given a policy $\pi \in \Pi^H$ and a configuration $p \in \mathcal{P}$, for every $(s,a) \in \mathcal{S} \times \mathcal{A}$ and $h \in [H]$ we define the configurator and agent Q-values as:

$$Q_{c,h}^{\pi,p}(s,a) = r_c(s) + \mathop{\mathbb{E}}_{s_{h'} \sim p, \pi} \left[ \sum_{h'=h+1}^{H} r_c(s_{h'}) | s_h = s, a_h = a \right],$$

$$Q_{o,h}^{\pi,p}(s,a) = r_o(s) + \mathop{\mathbb{E}}_{s_{h'} \sim p, \pi} \left[ \sum_{h'=h+1}^{H} r_o(s_{h'}) | s_h = s, a_h = a \right].$$

We denote with $V_{c,h}^{\pi,p}(s) = \mathbb{E}_{a \sim \pi_h(s)}[Q_{c,h}^{\pi,p}(s,a)]$ and $V_{o,h}^{\pi,p} = \mathbb{E}_{a \sim \pi_h(s)}[Q_{o,h}^{\pi,p}(s,a)]$ the value functions and with $V_c^{\pi,p} = \mathbb{E}_{s \sim \mu}[V_{c,1}^{\pi,p}(s)]$ and $V_o^{\pi,p} = \mathbb{E}_{s \sim \mu}[V_{o,1}^{\pi,p}(s)]$ the expected returns for the configurator and the agent respectively.

# 4 Problem Formulation

While for classical Conf-MDPs [29, 27] a notion of optimality is straightforward as agent and configurator share the same objective, in an NConf-MDP, they can display possibly conflicting interests. We assume a *sequential* interaction between the configurator and the agent that resembles the leader-follower protocol [10, 6, 34, 38]. First, the configurator (leader) selects an environment configuration $p \in \mathcal{P}$, where $\mathcal{P}$ is a finite set made of $M$ stochastic transition models $\mathcal{P} = \{p_1, \ldots, p_M\}$. Then the agent (follower) plays a policy chosen by a *best response function* $f : \mathcal{P} \to \Pi^H$, such that: $f(p) \in \arg\max_{\pi \in \Pi^H} V_o^{\pi,p}$. The solution concept that we use is the well-known *Stackelberg equilibrium* [43, 15, 30, 32, 19]. It captures the outcome in which the configurator's transition model is optimal, under the assumption that the agent will always respond optimally [26]. However, this definition includes the possibility of ties, i.e., situations in which multiple agent optimal policies exist, with possibly different performance for the configurator. Therefore, it is necessary to employ a *tie-breaking rule*, i.e., a criterion to select *one* agent best response. Different tie-breaking rules lead to different Stackelberg equilibria, and the two prevailing solution concepts in the literature are the *Strong Stackelberg Equilibrium* (SSE) and the *Weak Stackelberg Equilibrium* (WSE). A policy-transition model pair $(\pi^*, p^*)$ forms an SSE if ties are broken in favor of the configurator:

$$p^* \in \arg\max_{p \in \mathcal{P}} V_c^{f^S(p),p} \qquad \text{and} \qquad \pi^* := f^S(p) \in \arg\max_{\pi \in f(p)} V_o^{\pi,p}.$$

The WSE can be constructed by breaking the ties against the configurator. In the rest of the paper, we employ the concept of SSE; however, every result can be applied to any deterministic tie-breaking rule. We call $\pi_p^*$ the application of the best response function $f^S$ to a transition model $p$. Notice that the goal of the configurator is well-defined, whenever deciding the function $f^S$. From an online learning perspective, this goal is to minimize the expected regret:

$$\mathbb{E}[\text{Regret}(K)] = \mathbb{E}\left[ \sum_{k \in [K]} \max_{p \in \mathcal{P}} V_c^{\pi_p,p} - V_c^{\pi_{p_k},p_k} \right]. \tag{1}$$

To lighten the notation, in the following, we will denote with $\pi_i$ the agent's best response policy to the configuration $p_i$, i.e., $\pi_{p_i}^*$ and with $V^i$ the configurator expected returned attained with configuration $p_i$ and policy $\pi_i$, i.e., $V_c^{\pi_i,p_i}$. Finally, we denote with $V^* = \max_{i \in [M]} V^i$.

**Agent's Feedback**  The configurator knows its reward $r_c$, but it does not know the agent reward $r_o$. At each episode $k \in [K]$, the configurator selects a configuration $p_{I_k} \in \mathcal{P}$ and observes a trajectory of $H$ steps generated by the agent's best response policy $\pi_{I_k}$. We study two types of feedback:

- *Action-feedback* (Af). The configurator observes the states and the actions played by the agent $(s_1, a_1, \ldots, s_{H-1}, a_{H-1}, s_H)$, where $a_h \sim \pi_{I_k,h}(s_h)$.
- *Reward-feedback* (Rf). The configurator observes the states, the actions played by the agent, and a noisy feedback of the agent reward function $(s_1, \widetilde{r}_1, a_1, \ldots, s_{H-1}, \widetilde{r}_{H-1}, a_{H-1}, s_H, \widetilde{r}_H)$, where $a_h \sim \pi_{I_k,h}(s_h)$ and $\widetilde{r}_h$ is sampled from a distribution with mean $r_o(s)$ and support $[0,1]$.[2]

The Rf models situations in which the agent's reward is known under uncertainty, or it is obtained in an approximate way through Inverse Reinforcement Learning [33].

**Connections with Bandit Algorithms**  The online problem that we are facing can be seen as a stochastic multi-armed bandit [25], in which the arms are configurations, and the configurator receives a random realization of its expected return at every episode. Thus, in principle, it can be solved by standard algorithms for bandit problems, such as UCB1 [1]. These algorithms are computationally less demanding than those we will present in the next sections. On the other hand, they suffer regret that grows logarithmically, i.e., indefinitely, with the number of episodes. Indeed, they do not exploit either the information regarding the agent's policy or the structure induced by the agent's reward function. We will prove that, instead, the proposed algorithms, which use the problem structure, suffer bounded regret. Furthermore, our algorithms are combined with UCB1 confidence intervals, so their regret, at finite time, is never worse than the one of UCB1.

# 5  Optimistic Configuration Learning

In this section, we present two algorithms for the online learning problem introduced in Section 4. The first algorithm uses only the collected agent decisions to optimistically learn the best configuration (Section 5.1). In the second algorithm, we also use the noisy reward feedback to construct an algorithm that leverages the structure that links together all the transition probability models: the agent's reward function $r_o$ (Section 5.2). In Appendix C, we provide some hints about the adversarial case to illustrate the additional complexities that arise. In the adversarial setting, the agent can play a different policy at each step, inside the set of possible policies that satisfy the SSE.

## 5.1  Action-feedback Optimistic Configuration Learning

We start with the action-feedback (Af) setting, in which the configurator observes the agent's actions only. The idea at the basis of the algorithm we propose, *Action-feedback Optimistic Configuration Learning* (AfOCL), is to maintain, for each configuration, a set of *plausible* policies that contains an agent's best response policy. The configurator plays the transition model that maximizes an optimistic approximation of its value function. Specifically, for every $i \in [M]$, $k \in [K]$, and $h \in [H]$ we denote with $\mathcal{A}^i_{k,h}(s) \subseteq \mathcal{A}$ the set of plausible actions in state $s$ at step $h$ for configuration $p_i$ at the beginning of episode $k$. For every model $p_i$, the first time we visit an $(s,h)$-pair and observe the agent's action $a \sim \pi_{i,h}(\cdot|s)$, we set $\mathcal{A}^i_{k,h}(s) = \{a\}$. For the non-visited $(s,h)$-pairs, we leave $\mathcal{A}^i_{k,h}(s) = \mathcal{A}$. Based on this, we can compute an optimistic approximation $\widetilde{V}^i_{k,h}$ of the configurator value function $V^i_h$:

$$\widetilde{V}^i_{k,h}(s) = r_c(s) + \max_{a \in \mathcal{A}^i_{k,h}(s)} \sum_{s' \in \mathcal{S}} p_i(s'|s,a) \widetilde{V}^i_{k,h+1}(s'), \tag{2}$$

and $\widetilde{V}^i_{k,H}(s) = r_c(s)$. $\widetilde{V}^i_{k,h}$ can be computed applying a value-iteration-like algorithm [35] that employs the iterate as in Equation (2).[3] Clearly, if the agent is playing deterministically, it holds that $\mathcal{A}^i_{k,h}(s) = \{\pi_{i,h}(s)\}$ for all visited $(s,h)$-pairs and, consequently, $\widetilde{V}^i_{k,h}(s) \geq V^i_h(s)$. Instead, if the agent is playing stochastically, we possibly observe different actions whenever visiting $(s,h)$ and we record *just* the first one. The following lemma shows that even for stochastic agents, if the SSE tie-breaking rule is employed, $\widetilde{V}^i_{k,h}$ is optimistic.

---

[2]Clearly, the results we present can be directly extended to subgaussian distributions on the reward.

[3]Notice that the computational complexity decreases as the number of visited states increases and, in any case, is bounded by that of value iteration $\mathcal{O}\left(HS^2A\right)$. Therefore, the time complexity of AfOCL is $\mathcal{O}\left(KMHS^2A\right)$.

---

**Algorithm 1** Action-feedback Optimistic Configuration Learning (AfOCL).

1: **Input:** $\mathcal{S}, \mathcal{A}, H, \mathcal{P} = \{p_1, \ldots, p_M\}$
2: Initialize $\mathcal{A}_{1,h}^i(s) = \mathcal{A}$ for all $s \in \mathcal{S}$, $h \in [H]$, and $i \in [M]$
3: **for** episodes $1, 2, \ldots, K$ **do**
4:     Compute $\widetilde{V}_k^{i,\mathrm{UCB}}$ for all $i \in [M]$
5:     Compute $\widetilde{V}_k^i$ for all $i \in [M]$
6:     Play $p_{I_k}$ with $I_k \in \arg\max_{i \in [M]} \min\{\widetilde{V}_k^i, \widetilde{V}_k^{\mathrm{UCB}}\}$
7:     Observe $(s_{k,1}, a_{k,1}, \ldots, s_{k,H-1}, a_{k,H-1}, s_{k,H})$
8:     Compute the plausible actions for all $s \in \mathcal{S}$ and $h \in [H]$:

$$\mathcal{A}_{k+1,h}^i(s) = \begin{cases} \{a_{k,h}\} & \text{if } i = I_k \text{ and } s = s_{k,h} \text{ and } N_{k,h}(s) = 0 \\ \mathcal{A}_{k,h}^i(s) & \text{otherwise} \end{cases}$$

9: **end for**

---

**Lemma 5.1.** *The value function $\widetilde{V}_{k,h}^i(s)$ computed as in Equation (2) is such that $\widetilde{V}_{k,h}^i(s) \geq V_h^i(s)$ for all $s \in \mathcal{S}$, $h \in [H]$, and $i \in [M]$.*

In addition, we compute the confidence interval for UCB1 looking at the transition models as arms: $\widetilde{V}_k^{i,\mathrm{UCB}} = \bar{V}_k^i + H\sqrt{2\log k / N_{i,k}}$, where $\bar{V}_k^i$ is the sample mean of the observed return for model $p_i$ and $N_{i,k}$ is the number of times the algorithm plays model $i$ up to episode $k$. Thus, at each episode $k \in [K]$ the configurator plays the transition model $p_{I_k}$ maximizing the optimistic approximation:

$$I_k \in \arg\max_{i \in [M]} \min\{\widetilde{V}_k^i, \widetilde{V}_k^{i,\mathrm{UCB}}\}.$$

The pseudocode of AfOCL is reported in Algorithm 1.

**Regret Guarantees** We now provide an expected regret bound for the AfOCL algorithm. If the agent's policy $\pi_i$ is deterministic, it is not hard to get convinced that AfOCL suffers bounded regret since whenever an $(s, h)$-pair is visited under a $p_i$, the agent reveals its (deterministic) policy $\pi_i$. Thus, either an $(s, h)$-pair is visited with high probability, or it will impact only marginally on the performance. The main challenge arises when the agent is playing a stochastic policy $\pi_i$ for some $p_i$. AfOCL just memorizes the first observed action for each $(s, h)$, pretending the agent's policy to be deterministic. Let $\widehat{\pi}_i$ be the policy that plays the action memorized by AfOCL at the end of the $K$ episodes, filled with the true agent's policy for the non-visited $(s, h)$-pairs. By construction, the support of $\widehat{\pi}_i$ is contained into the support of the true agent's policy $\pi_i$. Clearly, if $\pi_i$ is optimal for the agent reward, $\widehat{\pi}_i$ is too. Furthermore, since the agent and the configurator are playing an SSE, $\widehat{\pi}_i$ will lead to the same configurator's performance as $\pi_i$. Indeed, if this were not the case, there would exist another deterministic policy optimal for the agent, leading to higher performance for the configurator, contradicting the definition of SSE. The following result shows that by switching $\pi_i$ with $\widehat{\pi}_i$ changes the regret just by a multiplicative factor depending on the mismatch between the visitation distributions induced by the two policies, $d_{i,h}$ and $\widehat{d}_{i,h}$ respectively.

**Theorem 5.1** (Regret of AfOCL). *Let $\mathcal{NCM} = (\mathcal{S}, \mathcal{A}, \mathcal{P}, \mu, r_c, r_o, H)$ with $\mathcal{P} = \{p_1, \ldots, p_M\}$ be the $M$ configurations. The expected regret of AfOCL at every episode $K > 0$ is bounded by:*

$$\mathbb{E}[Regret(K)] \leq \mathcal{O}\left( \min\left\{ \underbrace{H^2 \sum_{i \in [M]:\Delta_i > 0} \frac{\log(K)}{\Delta_i}}_{\text{UCB1 regret}} , \underbrace{MH^3 S^2 \rho}_{\text{AfOCL regret}} \right\} \right), \qquad (3)$$

*where $\rho$ is the $\max_{i \in [M]:\Delta_i > 0} \mathbb{E}\left[ \max_{s \in \mathcal{S}} \max_{h \in [H]} \frac{\widehat{d}_{i,h}(s)}{d_{i,h}(s)} \right]$.*

The result might be surprising as the regret is constant and independent of the suboptimality gaps between the configurations, i.e., $\Delta_i = V^* - V^i$ for every $i \in [M]$. As supported by intuition, we need to spend more time discarding MDPs that are more similar in performance to the optimal one. Formally, the maximum number of times a suboptimal configuration $p_i$ is played is proportional to $1/\Delta_i$ (and not proportional to $1/\Delta_i^2$ as in standard bandits). This is because we *just* need *one* visit to

every reachable state. We underline that the term $\rho$, which indicates the expected ratio between the estimated policy's induced states distribution and real policy's induced states distribution, is equal to 1 when the agent plays a deterministic policy and bounded by $SH$ in the worst case (see Lemma B.3). As far as we know, Theorem 5.1 is the *first problem-dependent* result for regret minimization for a multi-entity MDP. More details on the proof are given in the Appendix B.

## 5.2 Reward-feedback Optimistic Configuration Learning

The main drawback of AfOCL is that every transition model is treated separately, preventing from employing the underlying *structure* of the environment, which is represented by the agent reward function $r_o$. Indeed, if the configurator knew $r_o$, it could find the optimal configuration with no need for interaction by simply computing an agent's best response policies for the SSE.

The algorithm we propose in this section, *Reward-feedback Optimistic Configuration Learning* (RfOCL), employs the reward feedback (Rf), i.e., at every interaction, the configurator can see also a noisy version of the agent's reward function. The crucial point is that $r_o$ is the same regardless of the chosen configuration, and, for this reason, it provides a link between them. Specifically, for every $k \in [K]$ and $s \in \mathcal{S}$, RfOCL maintains a confidence interval for the agent reward function $\mathcal{R}_k(s) = [\underline{r}_{o,k}(s), \overline{r}_{o,k}(s)]$ obtained using the samples collected up to episode $k-1$ *regardless* of the played configuration. We apply Höeffding's inequality to build the confidence interval: $\widehat{r}_{o,k}(s) \pm \sqrt{\frac{\log(2SHk^2)}{\max\{N_k(s),1\}}}$, where $N_k(s)$ is the number of visits of state $s$ in the first $k-1$ episodes, and $\widehat{r}_{o,k}(s)$ is the sample mean of the observed rewards for state $s$ up to episode $k$. Given the estimated reward, for every configuration $i \in [M]$, we can compute a confidence interval for the agent's Q-values $\mathcal{Q}_{k,h}(s,a) = [\underline{Q}^i_{o,k,h}(s,a), \overline{Q}^i_{o,k,h}(s,a)]$, by simply applying the Bellman equation:

$$\underline{Q}^i_{o,k,h}(s,a) = \underline{r}_{o,k}(s) + \sum_{s' \in \mathcal{S}} p_i(s'|s,a) \max_{a' \in \mathcal{A}} \underline{Q}^i_{o,k,h+1}(s',a'),$$

$$\overline{Q}^i_{o,k,h}(s,a) = \overline{r}_{o,k}(s) + \sum_{s' \in \mathcal{S}} p_i(s'|s,a) \max_{a' \in \mathcal{A}} \overline{Q}^i_{o,k,h+1}(s',a'),$$

and $\underline{Q}^i_{o,k,H}(s,a) = \underline{r}_{o,k}(s)$ and $\overline{Q}^i_{o,k,H}(s,a) = \overline{r}_{o,k}(s)$. If the true reward function belongs to the confidence interval, i.e., $r_o \in \mathcal{R}_k$, then the true Q-value belongs to the corresponding confidence interval, i.e., $Q^i_h \in \mathcal{Q}_{k,h}$. Consequently, we can use $\mathcal{Q}_{k,h}$ to restrict the set of plausible actions in a state *without* actually observing the agent playing the action in that state. Indeed, the plausible actions are those that have a Q-value upper bound larger than the maximum Q-value lower bound:

$$\widetilde{\mathcal{A}}^i_{k,h}(s) = \left\{ a \in \mathcal{A} : \overline{Q}^i_{o,k,h}(s,a) \geq \max_{a' \in \mathcal{A}} \underline{Q}^i_{o,k,h}(s,a') \right\}. \tag{4}$$

In other words, if the upper Q-value of an action is smaller than the largest lower Q-value, it cannot be the greedy action, and it is discarded. Clearly, if we observe, for the first time, the agent playing an action in $(s,h)$ at episode $k$ we can reduce the plausible actions to the singleton $a_{k,h}$, as in the action-feedback setting (Section 5.1). Based on this refined definition of plausible actions, we can compute the optimistic estimate $\widetilde{V}^i_{k,h}$ of the configurator value function $V^i_h$ as in Equation (2) and proceed playing the optimistic configuration.

The pseudocode of RfOCL is reported in Algorithm 2. It is worth noting that we need to keep track of the states that have been already visited because for those, we know the agent's action, and there is no need to apply Equation (4). This is why we introduce the counts $N_{k,h}(s)$[4].

**Regret Guarantees**   We now give a regret bound for the RfOCL algorithm. Obviously, the same arguments for AfOCL can also be applied for this extended version, and then the regret bound of Theorem 5.1 is valid for RfOCL. Moreover, for this algorithm, we prove that the regret, under the following assumption, does not depend on the number of configurations.

**Assumption 1.** *There exists $\epsilon > 0$ such that:* $\min_{i \in [M]} \min_{s \in \mathcal{S}} \max_{h \in [H]} d^i_h(s) \geq \epsilon$, *where $d^i_h(s)$ is the probability of visiting the state $s \in \mathcal{S}$ at time $h \in [H]$ in configuration $p_i$ under the agent's best response policy $\pi_i$.*

---

[4]The value iteration dominates the computational complexity of an individual iteration of RfOCL (steps 5 and 9), leading, as for AfOCL, to $\mathcal{O}\left(KMHS^2A\right)$.

---

**Algorithm 2** Reward-feedback Optimistic Configuration Learning (RfOCL)

---

1: **Input:** $\mathcal{S}$, $\mathcal{A}$, $H$, $\mathcal{P} = \{p_1, \ldots, p_M\}$
2: Initialize $\mathcal{A}_{1,h}^i(s) = \mathcal{A}$ for all $s \in \mathcal{S}$, $h \in [H]$, and $i \in [M]$
3: Initialize $\overline{r}_{o,1}(s) = 1$, $\underline{r}_{o,1}(s) = 0$, and $N_{1,h}(s) = 0$ for all $s \in \mathcal{S}$ and $h \in [H]$
4: **for** episodes $1, 2, \ldots, K$ **do**
5:     Compute $\widetilde{V}_k^{i,\text{UCB}}$ for all $i \in [M]$
6:     Compute $\widetilde{V}_k^i$ for all $i \in [M]$
7:     Play $p_{I_k}$ with $I_k \in \arg\max_{i \in [M]} \min\{\widetilde{V}_k^i, \widetilde{V}_k^{\text{UCB}}\}$
8:     Observe $(s_{k,1}, \widetilde{r}_{k,1}, a_{k,1}, \ldots, s_{k,H-1}, \widetilde{r}_{k,H-1}, a_{k,H-1}, s_{k,H}, \widetilde{r}_{k,H})$
9:     Compute $\overline{r}_{o,k+1}(s)$, $\underline{r}_{o,k+1}(s)$, and $N_{k+1,h}(s)$ for all $s \in \mathcal{S}$ and $h \in [H]$ using $\widetilde{r}_{k,1} \cdots \widetilde{r}_{k,H}$
10:    Compute $\underline{Q}_{o,k+1,h}^i(s,a)$ and $\overline{Q}_{o,k+1,h}^i(s,a)$ for all $s \in \mathcal{S}$, $a \in \mathcal{A}$, $h \in [H]$, and $i \in [M]$
11:    Compute the plausible actions for all $s \in \mathcal{S}$ and $h \in [H]$:

$$\mathcal{A}_{k+1,h}^i(s) = \begin{cases} \{a_{k,h}\} & \text{if } i = I_k \text{ and } s = s_{k,h} \text{ and } N_{k,h}(s) = 0 \\ \mathcal{A}_{k,h}^i(s) & \text{if } N_{k,h}(s) > 0 \\ \widetilde{\mathcal{A}}_{k+1,h}^i(s) & \text{otherwise} \end{cases}$$

    with $\widetilde{\mathcal{A}}_{k+1,h}^i(s)$ as in Equation (4).
12: **end for**

---

This assumption requires that in every model $p_i \in \mathcal{P}$ the agent has non-zero probability, in some step $h$, to visit every state $s$. This allows shrinking the confidence intervals for the reward of every state to estimate the agent's policy correctly, regardless of the played configuration. Notice that this assumption is less strict than requiring the well-known ergodicity of the Markov process induced by *any* policy, used in many algorithms [9, 21, 44].[5] Under Assumption 1 we prove the following regret guarantee.

**Theorem 5.2** (Regret of RfOCL). *Let $\mathcal{NCM} = (\mathcal{S}, \mathcal{A}, \mathcal{P}, \mu, r_c, r_o, H)$ with $\mathcal{P} = \{p_1, \ldots, p_M\}$ be the $M$ configurations. Under Assumption 1, the expected regret of RfOCL at every episode $K > 0$ is bounded by:*

$$\mathbb{E}[\textit{Regret}(K)] \leq \mathcal{O}\left( \min\left\{ \underbrace{H^2 \sum_{i \in [M]:\Delta_i > 0} \frac{\log(K)}{\Delta_i}}_{\textit{UCB1 regret}} , \underbrace{MH^3S^2\rho}_{\textit{AfOCL regret}} , \underbrace{\overline{K}\Delta + \frac{\pi^2}{3}}_{\textit{RfOCL regret}} \right\} \right),$$

*where $\rho$ is defined as in Theorem 5.1, $\overline{K}$ is the smallest integer solution of the inequality $\overline{K} \geq 1 + \left( \frac{2H^2S^2 \log(2SH\overline{K}^2)}{2\Delta_Q^2} + \sqrt{\frac{\overline{K}-1}{2} \log(SH\overline{K}^2)} \right) \frac{1}{\epsilon}$, $\Delta = \max_{i \in [M]} \Delta_i$, i.e., the maximum suboptimality gap, and $\Delta_Q$ is the minimum positive gap of the agent's Q-values (see Appendix B).*

The regret bound removes the dependence on the number of models $M$, as $\overline{K}$ is clearly independent of $M$, but it introduces, as expected, a dependence on the minimum visitation probability $\epsilon$. The proof of the result is reported in Appendix B. Since RfOCL exploits additional information compared to AfOCL and the set of plausible actions $\mathcal{A}_{k,h}^i$ of RfOCL are subsets of those of AfOCL, the regret bound AfOCL (Theorem 5.1) also holds for RfOCL. Thus, we can take as regret bound for RfOCL the minimum between $\overline{K}\Delta + \frac{\pi^2}{3}$ and $MH^3S^2$. We underline that, as far as we know, this is the first proof that takes into consideration the *sub-optimality* gap of the uncontrollable entity, the agent, and the *sub-optimality* gap of the controllable entity, the configurator. This permits to derive a *problem dependent* regret bound. We think that similar techniques can also be of interest for Markov games.

## 6 Related Works

The idea of altering the environment dynamics to improve the agent's learning experience has been exploited before the introduction of Conf-MDPs. *Curriculum learning* [8] provides the agent with

---

[5]Moreover, the configurator can force this assumption since it has the *control* over the environmental transition model.

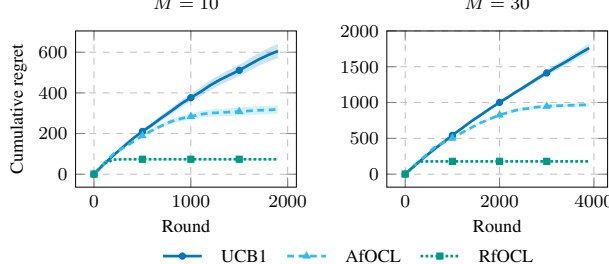

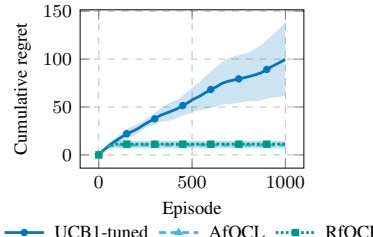

Figure 1: Cumulative regret for the Gridworld experiment. 50 runs, 98% c.i.

Figure 2: Cumulative regret for the Gridworld experiment without ergodicity. 50 runs, 98% c.i.

a sequence of environments, of increasing difficulty, to shape the learning process with possible benefits on the learning speed [e.g., 14, 16]. Although the learning process is carried out in a different environment, the configuration is typically performed in simulation only. The setting more similar to Non-Conf-MDP is the one presented in [47], where the configurator and the agent have opposite reward functions (similar to a zero-sum game).

In the Conf-MDP framework, instead, the configuration opportunities are an *intrinsic* property of the environment [29]. The initial approaches entitled the agent of the configuration activity and, consequently, this task was totally auxiliary to its learning experience [29, 39, 27]. More recently, it has been observed that environment configuration can be actuated even by an external entity, opening new opportunities for the application of environment configurability, including settings in which the configurator's interest conflicts with those of the agent. For instance, in [28] the configurator acts on the environment to induce the agent to reveal its capabilities in terms of perception and actuation. Instead, in [17] a threatener entity can change the transition probabilities either in a stochastic or adversarial manner. More generally, environment configuration carried out by an external entity has been studied in the field of planning as a form of *environment design* [48]. Thus, our NConf-MDP unifies these settings, allowing for arbitrary agent's and configurator's reward functions. An interesting connection is established with the *robust control* literature [31, 20]. Whenever the two reward functions are opposite, i.e., the interaction between the agent and the configuration is fully *competitive*, the resulting equilibrium corresponds to a robust policy. Indeed, while the agent tries to maximize its expected return, the configurator places the agent in the worst possible environment.

Configurable environments (cooperative and non-cooperative) share similarities with *environment design* [49]. At a high level, the two frameworks share analogous objectives: they both aim at determining an environment with a certain goal that can differ from that of the agent. However, there are some notable differences. In particular, the classical environment design formulation [49] assumes that the configurator (called "interested party") knows the agent's best response function, while in our approach, we learn it by interaction. Nevertheless, the general environment design makes no assumption about the underlying environment, that might not me an MDP. Instead, [22] limit to MDPs and considers a form of cooperative environment design in which the goal is to maximize the agent's performance. Interestingly, some works [22, 37] also account for a cost function to penalize expensive environment configurations.

The design of our approaches is based on the OFU principle used for stochastic multi-armed bandits [e.g., 23, 1, 18, 25] and MDPs [e.g., 2, 7, 3]. Moreover, our learning setting with reward feedback is related to structured bandits or bandits with correlated arms.[6] Interestingly, for certain structures, it is known that bounded regret is achievable [11, 24], a property that is enjoyed by both our algorithms. Our setting is also close to the Stochastic Games model, in which two or more agents act in an MDP to maximize their own reward functions. Recently, the stochastic game's framework gains growing interest [5, 4, 50], especially in the offline setting i.e., we can control all the agents. For this reason, these approaches do not apply to our setting, where we have the control of the configurator only.

---

[6]In our case, playing a single configuration provides information about the opponent's reward, which, in turn, provides information about the value of all configurations.

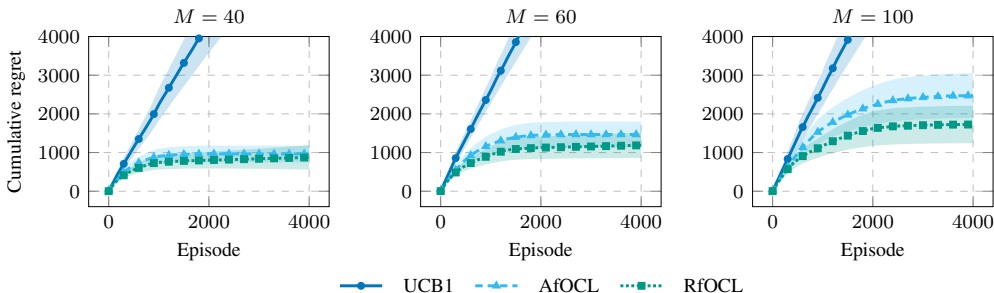

Figure 3: Cumulative regret as a function of the episodes for the Student-Teacher experiment. 50 runs, 98% c.i.

Although some works tackle the online setting [44, 45, 41], where we can control only one agent, all of these algorithms work in the zero-sum setting only.

# 7 Experiments

In this section, we provide the experimental evaluation of our algorithms in two different settings: when the policies are stochastic and when the policies are deterministic. For these experiments, we provide two novel environments: Configurable Gridworld and the Student-Teacher. We compare the algorithms with the standard (theoretical) implementation of UCB1 [1]. The environment description and additional results can be found in Appendix D.

**Stochastic policies**   The Configurable Gridworld is a configurable version of a classic $3 \times 3$ Gridworld. The agent's starting state is in the cell $(0, 1)$, and its goal is to minimize the number of steps required to reach the exit located in the cell $(2, 1)$. The configurator takes reward 1 when the agent occupies the central cell $(1, 1)$ and 0 otherwise. In a classic Gridworld, the optimal policy would be trivial, as the agent would proceed straight to the exit. In this Configurable Gridworld, instead, the configurator can set the "power" $p$ of a stochastic obstacle located in the cell $(1, 1)$. When the agent is in that cell and performs action "go right" to reach the exit, it will hit the obstacle, and will remain in the same position with probability $p$. The configurator's goal is to tune this probability to keep the agent in the central cell for the maximum number of steps.

The $M$ configurations differ in the probability $p$ and are obtained by a regular discretization of $[0, 1]$. In the first experiment (Figure 1), we considered 10 and 30 configurations with a number of episodes $K = 2000$ and $K = 4000$ and horizon $H = 10$. For this experiment, the agent plays *optimal stochastic policies*. We can see that AfOCL and RfOCL suffer constant regret, whereas UCB1 displays a logarithmic regret, as expected. Specifically, RfOCL outperforms AfOCL and stops playing suboptimal configuration in less than 500 episodes in both cases. This can be explained because, being Assumption 1 fulfilled (in fact, the agent has the probability 0.1 of failing its action), RfOCL is able to exploit the underlying structure of the problem more effectively.

**Non-Ergodicity**   In Figure 2, we have only three configurations designed to induce an optimal agent's policy that generates a non-ergodic Markov chain. In this case, the optimal policies are deterministic, and we violate Assumption 1. For this reason, we observe that AfOCL and RfOCL display very similar behavior but still significantly better than UCB1.

**Deterministic policies: Student-Teacher**   The Student-Teacher environment models a simple interaction between a student and a teacher. There is a set of exercises, with a different level of *teacher hardness* and *student hardness* each. The teacher has to decide the optimal sequence of exercises in order to make the student acquire as much knowledge as possible. The student's goal is to maximize the number of exercises and to reduce the *hardness* of the proposed exercises. At each timestep, the student decides whether to answer the exercise or not. If it answers, it receives a reward equal to the level of "correctness" of the exercise, the teacher receives a reward corresponding to the level of exercise's "teacher hardness", and they end up to the next exercise. If the student does not

answer, the student and the teacher will receive $-1$, and with a probability of $0.7$, the next exercise will be easier to solve. In Figure 3, the results with $M \in \{40, 60, 100\}$ and horizon $H = 10$ are shown. The configurations represent the distribution over the next exercise, given a positive answer. In every run, we change the *student hardness* of the exercises. We observe that both AfOCL and RfOCL suffer significantly less regret compared to UCB1 and tend to converge to constant regret as expected. It is interesting to observe that, in line with our analysis, the gap between AfOCL and RfOCL appears more evident as the number of configurations grows.

## 8    Conclusions

In this paper, we have introduced an extension of the Conf-MDP framework to account for possible non-cooperative interaction between the agent and the configurator. We focused on an online learning problem in this new setting, proposing two regret minimization algorithms for identifying the best environment configuration within a finite set, based on the principle of optimism in the face of uncertainty. We proved that even when the agent's policy is stochastic, and the configurator observes the agent's actions, it is possible to achieve finite regret that depends linearly on the admissible number configurations. Furthermore, we illustrated that we can remove this dependence if the configurator observes a possibly noisy version of the agent's reward and under sufficient regularity conditions on the environment. This paper also gives interesting insights on the importance of properly exploiting the available *feedback* to construct efficient algorithms. Moreover, as far as we know, the ones we have presented are the first *problem-dependent* regret results for multi-entity MDPs. The experimental evaluation showed that our algorithms display a convergence speed significantly faster than UCB1, and RfOCL tends to outperform AfOCL thanks to the exploitation of the additional structure. Future research directions include a deeper analysis of the adversarial setting, as well as the application to inverse reinforcement learning.

### Limitations and Societal Impact

Methods that incentive the manipulation of users' behavior can have, generally speaking, a negative societal impact, when used, for instance, in a marketing campaign. Nevertheless, our work is mainly theoretical and, at the present level, can hardly be used in a malevolent way. Another relevant aspect is the cost of environment configuration. We are aware that reconfiguring the environment is an activity that typically leads to higher costs compared with policy learning. However, we did not consider this aspect in the formalization of the Non-Cooperative Conf-MDP since it would possibly make the problem more complex (like, for instance, when considering bandits with switching costs).

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
