# A Notation

| | |
|---|---|
| $\mathcal{S}$ | State space |
| $\mathcal{A}$ | Action space |
| $\mathcal{P}$ | Configuration space |
| $M$ | Configuration space size |
| $r_o$ | Agent's reward function |
| $r_c$ | Configurator's reward function |
| $\mu$ | Initial state distribution |
| $H$ | Horizon |
| $Q_{c,h}^{\pi,p}(s,a)$ | Configurator's Q-value with policy $\pi$ and configuration $p$ |
| $Q_{o,h}^{\pi,p}(s,a)$ | Agent's Q-value with policy $\pi$ and configuration $p$ |
| $V_{c,h}^{\pi,p}(s)$ | Configurator's value function with policy $\pi$ and configuration $p$ |
| $V_{o,h}^{\pi,p}(s)$ | Agent's value function with policy $\pi$ and configuration $p$ |
| $V_c^{\pi,p}$ | Configurator's expected return with policy $\pi$ and configuration $p$ |
| $V_o^{\pi,p}$ | Agent's expected return with policy $\pi$ and configuration $p$ |
| $\pi_i = \pi_{p_i}^*$ | Agent's best response to configuration $p_i$ |
| $V^i = V_c^{\pi_{p_i}^*,p_i}$ | Configurator's expected return with the agent's best response policy $\pi_{p_i}^*$ to configuration $p_i$ |
| $V^* = V_c^{\pi_{p_{i*}}^*,p_{i*}}$ | Configurator's expected return with the agent's best response policy $\pi_{p_i}^*$ to the best configuration $p_{i*}$ |
| $\widetilde{V}_k^i$ | Optimistic configurator's expected return for configuration $p_i$ at episode $k$ |
| $\widetilde{\pi}_{i,k}$ | Estimated agent's best response policy for configuration $p_i$ at episode $k$ |
| $\Delta_i = V^* - V^i$ | Suboptimality gap of the configuration $p_i$ |
| $K$ | Number of episodes |
| $N_i$ | Number of times the configuration $p_i$ is played |
| $N_k(s)$ | Number of visits of state $s$ before episode $k$ |
| $N_{k,h}^i(s)$ | Number of visits of state $s$ at step $h$ before episode $k$ with configuration $p_i$ |
| $\underline{r}_{o,k}(s)$ | Lower confidence value for the agent's reward |
| $\overline{r}_{o,k}(s)$ | Upper confidence value for the agent's reward |
| $\widehat{r}_{o,k}(s)$ | Sample mean of observed rewards |
| $\underline{Q}_{o,k,h}^i(s,a)$ | Lower confidence value of the agent's Q-function with configuration $p_i$ |
| $\overline{Q}_{o,k,h}^i(s,a)$ | Upper confidence value of the agent's Q-function with configuration $p_i$ |
| $\mathcal{A}_{k,h}^i(s)$ | Set of agent's plausible actions in state $s$ at step $h$ up to episode $k$ |
| $d_h^i(s)$ | Visitation probability the state $s$ at step $h$ with configuration $p_i$ under the agent's best response policy $\pi_i$ |
| $\widetilde{d}_h^i(s)$ | Visitation probability the state $s$ at step $h$ with configuration $p_i$ under the estimated agent's best response policy $\widetilde{\pi}_{i,k}$ |

# B Missing Proofs

In this appendix, we report the proofs of the results presented in the main paper.

## B.1 Proofs of Section 5.1

**Lemma 5.1.** *The value function $\widetilde{V}_{k,h}^i(s)$ computed as in Equation (2) is such that $\widetilde{V}_{k,h}^i(s) \geq V_h^i(s)$ for all $s \in \mathcal{S}$, $h \in [H]$, and $i \in [M]$.*

*Proof.* We will prove the lemma by induction. We define $N_{k,h}^i(s')$ the number of times the state $s$ is visited at step $h$ with the configuration $p_i \in \mathcal{P}$ up to episode $k-1$.

*Case base* = $\widetilde{V}_{k,H}^i(s) \geq V_{k,H}^i(s)$ In this case is proven since $\widetilde{V}_{k,H}^i(s) = V_{k,H}^i(s) = r_c(s)$.

**Induction step** We assume that $\widetilde{V}_{k,h+1}^i(s) \geq V_{k,h+1}^i(s)$ and we will prove that $\widetilde{V}_{k,h}^i(s) \geq V_{k,h}^i(s)$.

$$\widetilde{V}_{k,h}^i(s) = r_c(s) + \max_{a \in \mathcal{A}_{k,h}^i(s)} \sum_{s' \in S} p_i(s'|s,a)\widetilde{V}_{k,h+1}^i(s)$$

$$\geq r_c(s) + \max_{a \in \mathcal{A}_{k,h}^i(s)} \sum_{s' \in S} p_i(s'|s,a)V_{k,h+1}^i(s),$$

This is true for the induction hypothesis. Now there are two cases:

- $N_{k,h}^i(s) > 0$ i.e, we have already visited the state $s$ at step $h$. In this case $\mathcal{A}_{k,h}^i(s) = a$ where $a$ is the action that by the agent the last time we have seen state $s$ at step $h$. If the policy is deterministic then:

$$r_c(s) + \max_{a \in \mathcal{A}_{k,h}^i(s)} \sum_{s' \in S} p_i(s'|s,a)V_{k,h+1}^i(s) = r_c(s) + \sum_{s' \in S} p_i(s'|s,\pi_p(s))V_{k,h+1}^i(s).$$

  Instead, if the policy is stochastic:

$$r_c(s) + \sum_{s' \in S} p_i(s'|s,a)V_{k,h+1}^i(s) = r_c(s) + \sum_{a' \in \mathcal{A}} \sum_{s' \in S} p_i(s'|s,\pi_p(s,a'))V_{k,h+1}^i(s),$$

  since, otherwise, the agent's policy played when we have seen the realization of $a$ is not optimal and does not respect the SSE definition.

- $N_{k,h}^i(s) = 0$. In this case $\mathcal{A}_{k,h}^i(s) = \mathcal{A}$ then, clearly,:

$$r_c(s) + \sum_{s' \in S} \max_{a \in \mathcal{A}_{k,h}^i(s)} p_i(s'|s,a)V_{k,h+1}^i(s) \geq r_c(s) + \sum_{a \in \mathcal{A}_{k,h}^i(s)} \sum_{s' \in S} p_i(s'|s,a)\pi_p(s,a)V_{k,h+1}^i(s).$$

  Then the result follows.

$\square$

We denote with $\widetilde{d}_h^i(s)$ the visitation probability of visiting state $s$ at step $h$ under transition model $p_i$ and playing the estimated agent's best response policy $\widetilde{\pi}_{i,k}$ (we will omit the subscript $k$ in the following). Then we start by constructing the policy $\widehat{\pi}_i$ such that:

$$\widehat{\pi}_{i,h}(\cdot|s) = \begin{cases} \widetilde{\pi}_{i,h}(\cdot|s) & \text{if} \quad N_{K,h}^i(s) > 0 \\ \pi_{i,h}(\cdot|s) & \text{if} \quad N_{K,h}^i(s) = 0 \end{cases} \tag{5}$$

A simple extension of Lemma 5.1 proves that the policy is optimal $\widehat{\pi}_{i,h}$ (thanks to the SSE definition). We call $\widehat{V}^i$ the expected return of $\widehat{\pi}_i$, and, obviously, $\widehat{V}^i = V^i$.

The visitation probabilities satisfy the following equalities for all $h \geq 2$:

$$d_h^i(s) = \sum_{s' \in \mathcal{S}} p_i(s|s',\cdot)^T \pi_{i,h}(\cdot|s')d_{h-1}^i(s')$$

$$\widehat{d}_h^i(s) = \sum_{s' \in \mathcal{S}} p_i(s|s',\cdot)^T \widehat{\pi}_{i,h}(\cdot|s')\widehat{d}_{h-1}^i(s') \tag{6}$$

$$\widetilde{d}_h^i(s) = \sum_{s' \in \mathcal{S}} p_i(s|s',\cdot)^T \widetilde{\pi}_{i,h}(\cdot|s')\widetilde{d}_{h-1}^i(s') = \sum_{s' \in \mathcal{S}} p_i(s|s',\widetilde{\pi}_{i,h}(s))\widehat{d}_{h-1}^i(s'),$$

where for $\widetilde{d}_h^i(s)$, we exploited the fact that $\widetilde{\pi}_{i,h}$ is deterministic, and $d_1^i(s) = \widetilde{d}_1^i(s) = \widehat{d}_1^i(s) = \mu(s)$.

**Lemma B.1.** *For every episode $k \in [K]$ and configuration $p_i \in \mathcal{P}$, the difference between the optimistic expected return $\widetilde{V}_k^i$ and the true expected return $V^i$ is bounded by:*

$$\widetilde{V}_k^i - V^i \leq 2H\rho_i \sum_{s \in \mathcal{S}} \sum_{h=1}^{H-1} d_h^i(s) \mathbb{1}\left\{N_{k,h}^i(s) = 0\right\}. \tag{7}$$

*where $N_{k,h}^i(s)$ is the number of times the state $s \in \mathcal{S}$ is visited at step $h \in [H]$ with the configuration $p_i \in \mathcal{P}$ up to episode $k-1$, where $\rho_i = \max_{s \in \mathcal{S}} \max_{h \in H} \frac{\widehat{d}_{i,h}(s)}{d_{i,h}(s)}$.*

*Proof.* As observed above, $\pi_i$ and $\widehat{\pi}_i$, given the definition of SSE, induce the same value function $V^i = \widehat{V}^i$. Thus, we have

$$\widetilde{V}_k^i - V^i = \widetilde{V}_k^i - \widehat{V}^i = \sum_{s \in \mathcal{S}} \left[\mu(s)r(s) - \mu(s)r(s) + \sum_{h=2}^{H}(\widetilde{d}_h^i(s) - \widehat{d}_h^i(s))r(s)\right] \tag{P.1}$$

$$\leq \sum_{s \in \mathcal{S}} \sum_{h=2}^{H} \left|\widetilde{d}_h^i(s) - \widehat{d}_h^i(s)\right| \tag{P.2}$$

$$= \sum_{s \in \mathcal{S}} \sum_{h=1}^{H-1} \left|\sum_{s' \in \mathcal{S}} \widetilde{d}_h^i(s')p_i(s|s', \widetilde{\pi}_{i,h}(s')) - \widehat{d}_h^i(s')p_i(s|s', \cdot)^T \widehat{\pi}_{i,h}(\cdot|s')\right| \tag{P.3}$$

$$= \sum_{s \in \mathcal{S}} \sum_{h=1}^{H-1} \sum_{s' \in \mathcal{S}} \left|\widetilde{d}_h^i(s') - \widehat{d}_h^i(s')\right| p_i(s|s', \widetilde{\pi}_{i,h}(s')) + \widehat{d}_h^i(s') \left|p_i(s|s', \widetilde{\pi}_{i,h}(s')) - p_i(s|s', \cdot)^T \widehat{\pi}_{i,h}(\cdot|s')\right| \tag{P.4}$$

$$= \sum_{s' \in \mathcal{S}} \sum_{h=2}^{H-1} \left|\widetilde{d}_h^i(s') - \widehat{d}_h^i(s')\right| + \sum_{s \in \mathcal{S}} \sum_{s' \in \mathcal{S}} \sum_{h=1}^{H-1} \widehat{d}_h^i(s') \left|p_i(s|s', \widetilde{\pi}_{i,h}(s')) - p_i(s|s', \cdot)^T \widehat{\pi}_{i,h}(\cdot|s')\right|$$

$$= \sum_{H'=2}^{H} \sum_{s \in \mathcal{S}} \sum_{s' \in \mathcal{S}} \sum_{h=1}^{H'-1} \widehat{d}_h^i(s') \left|p_i(s|s', \widetilde{\pi}_{i,h}(s')) - p_i(s|s', \cdot)^T \widehat{\pi}_{i,h}(\cdot|s')\right| \tag{P.5}$$

$$\leq H \sum_{s' \in \mathcal{S}} \sum_{h=1}^{H-1} \widehat{d}_h^i(s') \sum_{s \in \mathcal{S}} \left|p_i(s|s', \widetilde{\pi}_{i,h}(s')) - p_i(s|s', \cdot)^T \widehat{\pi}_{i,h}(\cdot|s')\right| \tag{P.6}$$

$$\leq 2H \sum_{s' \in \mathcal{S}} \sum_{h=1}^{H-1} \mathbb{1}\left\{N_{k,h}^i(s) = 0\right\} \widehat{d}_h^i(s') \tag{P.7}$$

$$= 2H \sum_{s' \in \mathcal{S}} \sum_{h=1}^{H-1} \mathbb{1}\left\{N_{k,h}^i(s) = 0\right\} d_h^i(s') \frac{\widehat{d}_h^i(s')}{d_h^i(s')}, \tag{P.8}$$

where in line (P.1) we use the definition of expected return. In line (P.2) we bound the value of every reward with its maximum value 1. In line (P.3) we expanded the probability distribution of visiting states using Equations (6). In line (P.4) we observe that $\widetilde{d}_1^i(s') - \widehat{d}_1^i(s') = \mu(s) - \mu(s) = 0$ to make the first summation starting from $h = 2$. In line (P.5), we apply the recursion with line (P.2). In line (P.6), we bound $H' \leq H$ and observe that the outer summation has less than $H$ terms. Finally, in line (P.8) we upper bound the differences between the two probabilities with 2, and we use the fact that when we have seen a state $s$ at step $h$ with a configuration $p_i$ the two policies are equal by construction. $\square$

**Lemma B.2.** *A configuration $p_i \in \mathcal{P}$ is no longer played after episode $k \in [K]$ if for every state $s \in \mathcal{S}$ and $h \in [H]$, with $d_h^i(s) \geq \frac{\Delta_i - c}{2H^2 S \rho_i}$, we have $N_{k,h}^i(s) > 0$, where $c > 0$ is arbitrary and $\Delta_i = V^* - V^i$.*

*Proof.* It suffices to prove that the optimistic expected return satisfies $\widetilde{V}_k^i < V^*$, that, in turn, will satisfy $V^* \leq \widetilde{V}_k^{i^*}$ where $i^* \in \arg\max_{i \in [M]} V^i$ (this way configuration $i$ will no longer be played):

$$\widetilde{V}_k^i = V^i + \widetilde{V}_k^i - V^i$$

$$\leq V^i + 2H\rho_i \sum_{s \in \mathcal{S}} \sum_{h=1}^{H-2} d_h^i(s)\mathbb{1}\left\{N_{h,k}^i(s) = 0\right\} \tag{P.9}$$

$$\leq V^i + 2H^2 S\rho_i \frac{\Delta_i - c}{2H^2 S\rho_i} \tag{P.10}$$

$$= V^i + \Delta_i - c < V^*, \tag{P.11}$$

where in line (P.9) we apply Lemma B.1. In line (P.10) we bound the state visitation probabilities of the $(s, h)$ pairs with $N_{h,k}^i(s) > 0$ with their maximum value, as in the statement hypothesis. In line (P.11) we use the fact that $\Delta_i = V^* - V_i$. $\qquad\square$

**Theorem 5.1** (Regret of AfOCL). *Let $\mathcal{NCM} = (\mathcal{S}, \mathcal{A}, \mathcal{P}, \mu, r_c, r_o, H)$ with $\mathcal{P} = \{p_1, \ldots, p_M\}$ be the $M$ configurations. The expected regret of AfOCL at every episode $K > 0$ is bounded by:*

$$\mathbb{E}[\mathit{Regret}(K)] \leq \mathcal{O}\left( \min\left\{ H^2 \underbrace{\sum_{i \in [M]:\Delta_i > 0} \frac{\log(K)}{\Delta_i}}_{\text{UCB1 regret}} , \underbrace{MH^3 S^2 \rho}_{\text{AfOCL regret}} \right\} \right), \tag{3}$$

*where $\rho$ is the $\max_{i \in [M]:\Delta_i > 0} \mathbb{E}\left[ \max_{s \in \mathcal{S}} \max_{h \in [H]} \frac{\widehat{d}_{i,h}(s)}{d_{i,h}(s)} \right]$.*

*Proof.* We start by dividing the analysis between the UCB1 algorithm and the proposed new algorithm. For the UCB1 algorithm the regret is straightforward from [1]:

$$\mathbb{E}[\mathit{Regret}(K)] \leq \mathcal{O}\left( H^2 \sum_{i \in [M]:\Delta_i > 0} \frac{\log(K)}{\Delta_i} \right),$$

since the random variables $V^i$ for each model $i \in [M]$ have their support in $[0, H]$.

Then, we analyze the regret of the proposed algorithm. We rephrase the regret as:

$$\mathbb{E}[\mathit{Regret}(K)] = \sum_{i \in [M]:\Delta_i > 0} \Delta_i \mathbb{E}[N_i],$$

where $N_i$ is the number of times that the algorithm plays model $p_i$ which is not the optimal configuration $p_{i^*}$. We start bounding for every configuration $p_i$ s.t. $\Delta_i > 0$ the expected value of $N_i$. We denote with $k_l^i$ the round at which model $i$ is selected for the $l$-th time:

$$\mathbb{E}[N_i] \leq \sum_{l=0}^{K} \Pr(N_i \geq l)$$

$$\leq \sum_{l=0}^{\infty} \Pr(N_i \geq l) \tag{P.12}$$

$$\leq \sum_{l=0}^{\infty} \Pr\left( \widetilde{V}_{k_l^i}^i - V^* \geq 0 \right), \tag{P.13}$$

$$\tag{P.14}$$

where in line (P.12) we extend the sum to $\infty$. In line (P.13) we exploit the fact that if configuration $i$ is selected then it must be $\widetilde{V}_{k_l^i}^i \geq \widetilde{V}_{k_l^i}^{i^*}$ and, because of optimism $\widetilde{V}_{k_l^i}^{i^*} \geq V^*$. Then, we observe that for Lemma B.2, if configuration $i$ is played at time $k_l^i$, then there must exists $s \in \mathcal{S}$ and $h \in [H]$ with $d_h^i(s) \geq \frac{\Delta_i - c}{2H^2 S}$ that is not played yet. Formally:

$$\mathbb{E}[N_i] \leq \sum_{l=0}^{\infty} \Pr\left(\widetilde{V}_{k_l^i}^i - V^* \geq 0\right)$$

$$\leq 1 + \sum_{l=1}^{\infty} \Pr\left(\exists s \in \mathcal{S}, \exists h \in [H] : d_h^i(s) \geq \frac{\Delta_i - c}{2H^2 S \rho_i} \wedge N_{k_l^i, h}^i(s) = 0\right) \tag{P.15}$$

$$\leq 1 + \sum_{l=1}^{\infty} \sum_{s \in \mathcal{S}, h \in [H]} \Pr\left(d_h^i(s) \geq \frac{\Delta_i - c}{2H^2 S \rho_i} \wedge N_{k_l^i, h}^i(s) = 0\right) \tag{P.16}$$

$$\leq 1 + \sum_{l=1}^{\infty} \sum_{s \in \mathcal{S}, h \in [H]} \mathbb{E}\left[\Pr\left(N_{k_l^i, h}^i(s) = 0 \Big| d_h^i(s) \geq \frac{\Delta_i - c}{2H^2 S \rho_i}\right) \mathbb{1}\left\{d_h^i(s) \geq \frac{\Delta_i - c}{2H^2 S \rho_i}\right\}\right] \tag{P.17}$$

$$\leq 1 + \sum_{l=1}^{\infty} \sum_{s \in \mathcal{S}, h \in [H]} \mathbb{E}\left[\Pr\left(N_{k_l^i, h}^i(s) = 0 \Big| d_h^i(s) \geq \frac{\Delta_i - c}{2H^2 S \rho_i}\right)\right] \tag{P.18}$$

$$\leq 1 + SH \, \mathbb{E}\left[\sum_{l=1}^{\infty}\left(1 - \frac{\Delta_i - c}{2H^2 S \rho_i}\right)^{l-1}\right] \tag{P.19}$$

$$= 1 + SH \, \mathbb{E}\left[\frac{1}{\frac{\Delta_i - c}{2H^2 S \rho_i}}\right] \leq 1 + \frac{2H^3 S^2 \, \mathbb{E}[\rho_i]}{\Delta_i - c}, \tag{P.20}$$

where, in line (P.15) we use Lemma B.2. In line (P.16) we use the union bound over the set employed for existential quantification. In line (P.17) we employed the definition of conditional probability and in line (P.18) we bounded the indicator with 1. In line (P.19) we bound the probability as $\Pr\left(N_{k_l^i, h}^i(s) = 0\right) = (1 - d_h^i(s))^{l-1}$, thanks to the independence of the rounds. In line (P.20) we use the geometric series properties.

So the expected regret is bounded by:

$$\mathbb{E}[\text{Regret}(K)] = \sum_{i \in [M]: \Delta_i > 0} \Delta_i \, \mathbb{E}[N_i] \leq \sum_{i \in [M]: \Delta_i > 0} \Delta_i \left(\frac{2H^3 S^2 \, \mathbb{E}[\rho_i]}{\Delta_i - c} + 1\right) \leq 3MH^3 S^2 \rho,$$

having taken the infimum over $c > 0$ and $\rho = \max_{i \in [M]: \Delta_i > 0} \mathbb{E}[\rho_i]$. $\qquad \square$

**Lemma B.3.** *The expected value of $\frac{\widehat{d}_{i,h}(s)}{d_{i,h}(s)}$, taken w.r.t. the randomness of the episodes, is 1. Moreover, the expectation of $\rho_i = \max_{s \in \mathcal{S}} \max_{h \in [H]} \frac{\widehat{d}_{i,h}(s)}{d_{i,h}(s)}$, taken w.r.t. the randomness of the episodes, is bounded by $SH$.*

*Proof.* First of all, we observe that, given its definition, for every $s, s' \in \mathcal{S}$ and $h, h' \in [H]$ such that $(s, h) \neq (s', h')$ we have that $\widehat{\pi}_h(\cdot|s)$ and $\widehat{\pi}_{h'}(\cdot|s')$ are independent. This is because $\widehat{\pi}$ is a policy that plays deterministically an action in each $(s, h)$, selected by querying the true agent's policy $\pi$. Consequently, since actions played by the agent in different $(s, h)$ are independent, also the policy entries $\widehat{\pi}_h(\cdot|s)$ are independent for different $(s, h)$-pairs. Moreover, $\mathbb{E}[\widehat{\pi}_h(\cdot|s)] = \pi_h(\cdot|s)$, where the expectation is taken w.r.t. the randomness of the episodes. We are going to prove by induction that $\mathbb{E}\left[\widehat{d}_{i,h}(s)\right] = d_{i,h}(s)$. Let us consider the case $h = 2$:

$$\mathbb{E}\left[\widehat{d}_{i,2}(s)\right] = \sum_{s' \in S} \mu(s') p(s|s', \cdot)^T \mathbb{E}[\widehat{\pi}_{i,1}(\cdot|s')] = d_{i,1}(s).$$

By induction, suppose that the statement hold for all $h' \leq h$, we prove it for $h + 1$:

$$\mathbb{E}\left[\widehat{d}_{i,h+1}(s)\right] = \sum_{s' \in S} \mathbb{E}\left[\widehat{d}_{i,h}(s') p(s|s', \cdot)^T \widehat{\pi}_{i,h}(\cdot|s')\right]$$

$$= \sum_{s' \in S} \mathbb{E}\left[\widehat{d}_{i,h}(s')\right] p(s|s', \cdot)^T \mathbb{E}\left[\widehat{\pi}_{i,h}(\cdot|s')\right]$$

$$= \sum_{s' \in S} d_{i,h}(s') p(s|s', \cdot)^T \pi_{i,h}(\cdot|s') = d_{i,h+1}(s),$$

where the last but one line derives from the fact that $\widehat{d}_{i,h}$ and $\widehat{\pi}_{i,h}$ are independent. This is due to the fact that $\widehat{d}_{i,h}$ depends on the policies $\{\widehat{\pi}_{i,h'}\}$ for $h' < h$ only that, in turn, are independent from $\{\widehat{\pi}_{i,h'}\}$ as noted at the

beginning of the proof. The last line follows from the inductive hypothesis. For the second statement, we have:

$$\mathbb{E}[\rho_i] = \mathbb{E}\left[\max_{s \in \mathcal{S}} \max_{h \in [H]} \frac{\widehat{d}_{i,h}(s)}{d_{i,h}(s)}\right] \leq \sum_{s \in \mathcal{S}} \sum_{h \in [H]} \mathbb{E}\left[\frac{\widehat{d}_{i,h}(s)}{d_{i,h}(s)}\right] = SH,$$

having exploited the first statement. $\qquad\square$

## B.2  Proofs of Section 5.2

In this section, we are going to prove the regret bound RfOCL. In this second algorithm the configurator can observe at every episode also a realization of the agent's reward function. In the following we will show how the algorithm exploits this information under Assumption 1.

We start defining the good events $G_k$ for $k \in [K]$:

$$G_k = \left\{\forall s \in \mathcal{S}, |\widehat{r}_{o,k}(s) - r_o(s)| \leq \sqrt{\frac{\log(2SHk^2)}{2N_k(s)}}\right\}$$

The event $G_k$ means that, at episode $k \in [K]$, the estimated rewards of each state $s \in \mathcal{S}$ are inside the confidence intervals.

**Lemma B.4.** *For every configuration $p_i \in \mathcal{P}$ and state action pair $(s, a) \in \mathcal{S} \times \mathcal{A}$, the difference between the optimistic state-action value function $\overline{Q}^i_{o,k,1}(s, a)$ and the true optimal state-action value function $Q^i_{o,1}(s, a)$ is bounded by:*

$$\overline{Q}^i_{o,k,1}(s, a) - Q^i_{o,1}(s, a) \leq \overline{r}_{o,k}(s) - r_o(s) + \sum_{s' \in \mathcal{S}} \sum_{h=2}^{H} \overline{d}^i_{k,h}(s') \left(\overline{r}_{o,k}(s') - r_o(s')\right),$$

*where $\overline{d}^i_{k,h}$ the visitation distribution induced by a greedy policy $\overline{\pi}_{i,k}$ w.r.t. $\overline{Q}^i_{o,k}$. Similarly, the difference between the true optimal state-action value function $Q^i_{o,1}(s, a)$ and the pessimistic state-action value function $\underline{Q}^i_{o,k,1}(s, a)$ is bounded by:*

$$Q^i_{o,1}(s, a) - \underline{Q}^i_{o,k,1}(s, a) \leq r_o(s) - \underline{r}_{o,k}(s) + \sum_{s' \in \mathcal{S}} \sum_{h=2}^{H} d^i_{k,h}(s') \left(r_o(s') - \underline{r}_{o,k}(s')\right).$$

*Proof.* The proof is basically taken from [46, 3, 42]:

$$\overline{Q}^i_{o,k,1}(s, a) - Q^i_{o,1}(s, a) \leq \overline{Q}^i_{o,k,1}(s, a) - Q^{\overline{\pi}_{i,k}}_{o,1}(s, a) \tag{P.21}$$

$$= \overline{r}_{o,k}(s) - r_o(s) + \sum_{s' \in \mathcal{S}} \sum_{h=2}^{H} \overline{d}^i_{k,h}(s') \left(\overline{r}_{o,k}(s') - r_o(s')\right). \tag{P.22}$$

where line (P.21) is due to $Q^i_{o,1}(s, a) \geq Q^{\overline{\pi}_{i,k}}_{o,1}(s, a)$, recalling that $Q^i_{o,1}$ is the optimal Q-value for the agent, under configuration $p_i$ and the optimal agent's policy. Line (P.21) derives form the application of the simulation lemma since $\overline{Q}^i_{o,k,1}(s, a)$ and $Q^{\overline{\pi}_{i,k}}_{o,1}(s, a)$ are under the same policy $\overline{\pi}_{i,k}$. For the second statement, we proceed analogously by simply observing that $\underline{Q}^i_{o,k,1}(s, a) \geq \underline{Q}^{\pi_i}_{o,1}(s, a)$ where $\pi_i$ is a greedy policy w.r.t. $Q^i_{o,k}(s, a)$. $\qquad\square$

**Lemma B.5.** *If for all $k \in [K]$, the good events $G_k$ hold, for all state-action pairs $(s, a) \in \mathcal{S} \times \mathcal{A}$, $h \in [H]$, and configuration $p_i \in \mathcal{P}$ it holds that:*

$$\overline{Q}^i_{o,k,1}(s, a) - Q^i_{o,1}(s, a) \leq SH\sqrt{\frac{\log(2SHk^2)}{2N_k(s)}},$$

$$Q^i_{o,1}(s, a) - \underline{Q}^i_{o,k,1}(s, a) \leq SH\sqrt{\frac{\log(2SHk^2)}{2N_k(s)}}.$$

*Proof.* We apply Lemma B.4, recall that $\bar{r}_{o,k}(s) = \hat{r}_{o,k}(s) + \sqrt{\frac{\log(2SHk^2)}{2N_k(s)}}$ and $\underline{r}_{o,k}(s) = \hat{r}_{o,k}(s) - \sqrt{\frac{\log(2SHk^2)}{2N_k(s)}}$, and make use of the definition of the events $G_k$. Then, we bound the visitation distribution with 1. $\qquad\square$

**Lemma B.6.** *Let* $s \in \mathcal{S}$ *be a state with minimum visitation probability* $d(s) := \min_{i \in [M]} \max_{h \in [H]} d_h^i(s) > 0$. *Then, at episode* $k \in [K]$, *for every* $\delta_k \in (0,1)$, *with probability at least* $1 - \delta_k$ *it holds that:*

$$N_k(s) \geq (k-1)d(s) - \sqrt{\frac{k-1}{2}\log\left(\frac{1}{\delta_k}\right)}.$$

*Proof.* First of all, we define the random variable $N_k^u(s)$ as the count of the visits to state $s$, where multiple visits in the same episode are considered just once:

$$N_k^u(s) = \sum_{i=1}^{k-1} \mathbb{1}\left\{\exists h \in [H] \ : \ s_{k,h} = s\right\}.$$

Clearly, $N_k^u(s) \leq N_k(s)$ and, consequently, $\mathbb{E}[N_k^u(s)] \leq \mathbb{E}[N_k(s)]$. The expectation of $\mathbb{E}[N_k^u(s)]$ can be bounded as:

$$\mathbb{E}[N_k^u(s)] = \mathbb{E}\left[\sum_{i=1}^{k-1} \mathbb{1}\left\{\exists h \in [H] \ : \ s_{k,h} = s\right\}\right]$$

$$= \sum_{i=1}^{k-1} \Pr\left(\exists h \in [H] \ : \ s_{k,h} = s | p_{I_k}, \pi_{I_k}\right) \tag{P.23}$$

$$= \sum_{i=1}^{k-1} \Pr\left(\bigcup_{h \in [H]} \{s_{k,h} = s\} | p_{I_k}, \pi_{I_k}\right) \tag{P.24}$$

$$\geq \sum_{i=1}^{k-1} \max_{h \in [H]} \Pr\left(s_{k,h} = s | p_{I_k}, \pi_{I_k}\right) \tag{P.25}$$

$$= \sum_{i=1}^{k-1} \max_{h \in [H]} d_h^{I_k}(s) \tag{P.26}$$

$$\geq (k-1)\min_{i \in [M]} \max_{h \in [H]} d_h^i(s) = (k-1)d(s), \tag{P.27}$$

where line (P.23) and line (P.24) we simply rewrite the expectation as probability. In line (P.25) we bound the probability of the union with just one term. In line (P.26) we employ the definition of $d_h^{I_k}(s)$. Finally, in line (P.27), we take the minimum over $I_k$. Since $0 \leq N_k^u(s) \leq k-1$, by using Höeffding's inequality, we have that with probability at least $1 - \delta_k$ it holds that:

$$N_k^u(s) \geq \mathbb{E}[N_k^u(s)] - \sqrt{\frac{k-1}{2}\log\frac{1}{\delta_k}} \geq (k-1)d(s) - \sqrt{\frac{k-1}{2}\log\frac{1}{\delta_k}},$$

having used the lower bound on $\mathbb{E}[N_k^u(s)]$. The result follows from recalling that $\mathbb{E}[N_k^u(s)] \leq \mathbb{E}[N_k(s)]$. $\quad\square$

**Lemma B.7.** *If for all* $k \in [K]$, *the good events* $G_k$ *hold, and for all* $s \in \mathcal{S}$ *it holds that* $\sqrt{\frac{\log(2SHk^2)}{2N_k(s)}} \leq \frac{\Delta_Q - c}{2SH}$, *with arbitrary* $c > 0$, *then for every configuration* $p_i \in \mathcal{P}$ *we have that* $\tilde{\pi}_{i,k} = \pi_i$.

*Proof.* Let $\Delta_Q$ be the minimum gap between the Q-function in the optimal action and a different action in all transition probabilities $p_i \in \mathcal{P}$:

$$\Delta_Q = \min_{i \in [M]} \min_{s \in \mathcal{S}} \min_{h \in [H]} \left\{ \max_{a \in \mathcal{A}} Q_{o,h}^i(s,a) - \max_{a' \in \mathcal{A} \setminus \arg\max_{a \in \mathcal{A}} Q_{o,h}^i(s,a)} Q_{o,h}^i(s,a') \right\}.$$

For all $s \in \mathcal{S}$ and $h \in [H]$, we denote with $a^* = \arg\max_{a \in \mathcal{A}} Q^i_{o,h}(s,a)$ and we have for all $a \in \mathcal{A} \setminus \{a^*\}$:

$$\overline{Q}^i_{o,k,h}(s,a) - \underline{Q}^i_{o,k,h}(s,a^*) = \overline{Q}^i_{o,k,h}(s,a) - \underline{Q}^i_{o,k,h}(s,a^*) \pm Q^i_{o,h}(s,a) \pm Q^i_{o,h}(s,a^*)$$

$$= \underbrace{\overline{Q}^i_{o,k,h}(s,a) - Q^i_{o,h}(s,a)}_{(A)} + \underbrace{Q^i_{o,h}(s,a^*) - \underline{Q}^i_{o,k,h}(s,a^*)}_{(B)}$$

$$+ \underbrace{Q^i_{o,h}(s,a) - Q^i_{o,h}(s,a^*)}_{(C)}$$

$$\leq 2SH\sqrt{\frac{\log(2SHk^2)}{2N_k(s)}} - \Delta_Q$$

$$\leq 2SH\frac{\Delta_Q - c}{2SH} - \Delta_Q \leq -c,$$

where for (A) and (B) we applied Lemma B.5 and for (C) we used the definition of $\Delta_Q$. We have proved that the lower bound on the Q-value of the optimal action $\underline{Q}^i_{o,k,h}(s,a^*)$ falls above the upper bound on the Q-value of all other actions $\overline{Q}^i_{o,k,h}(s,a)$. Consequently, the greedy action will be properly identified and $\widetilde{\pi}_{i,k} = \pi_i$. □

**Theorem 5.2** (Regret of RfOCL). *Let $\mathcal{NCM} = (\mathcal{S}, \mathcal{A}, \mathcal{P}, \mu, r_c, r_o, H)$ with $\mathcal{P} = \{p_1, \ldots, p_M\}$ be the $M$ configurations. Under Assumption 1, the expected regret of RfOCL at every episode $K > 0$ is bounded by:*

$$\mathbb{E}[\text{Regret}(K)] \leq \mathcal{O}\left( \min\left\{ \underbrace{H^2 \sum_{i \in [M]: \Delta_i > 0} \frac{\log(K)}{\Delta_i}}_{\text{UCB1 regret}} , \underbrace{MH^3S^2\rho}_{\text{AfOCL regret}} , \underbrace{\overline{K}\Delta + \frac{\pi^2}{3}}_{\text{RfOCL regret}} \right\} \right),$$

*where $\rho$ is defined as in Theorem 5.1, $\overline{K}$ is the smallest integer solution of the inequality $\overline{K} \geq 1 + \left( \frac{2H^2S^2\log(2SH\overline{K}^2)}{2\Delta_Q^2} + \sqrt{\frac{\overline{K}-1}{2}\log(SH\overline{K}^2)} \right)\frac{1}{\epsilon}$, $\Delta = \max_{i \in [M]}\Delta_i$, i.e., the maximum suboptimality gap, and $\Delta_Q$ is the minimum positive gap of the agent's Q-values (see Appendix B).*

*Proof.* We rewrite the expected regret as follows:

$$\mathbb{E}[\text{Regret}(K)] = \sum_{k=1}^{K} \left( \mathbb{E}[\Delta_{I_k}\mathbb{1}\{G_k\}] + \mathbb{E}[\Delta_{I_k}\mathbb{1}\{\neg G_k\}] \right)$$

$$\leq \underbrace{\sum_{k=1}^{K} \mathbb{E}[\Delta_{I_k} | G_k]}_{(A)} + \underbrace{H\sum_{k=1}^{K} \Pr(\neg G_k)}_{(B)},$$

where we bounded $\Pr(G_k) \leq 1$ in term (A) and $\Delta_{I_k}$ with its maximum value $H$ in term (B). We start bounding the (B) term:

$$H\sum_{k=1}^{K}\Pr(\neg G_k) = H\sum_{k=1}^{K}\Pr\left( \exists s \in \mathcal{S} \text{ s.t. } |\widehat{r}_{o,k}(s) - r(s)| > \sqrt{\frac{\log(2SHk^2)}{2N_k(s)}} \right) \tag{P.28}$$

$$\leq H\sum_{k=1}^{K}\sum_{s \in \mathcal{S}}\Pr\left( |\widehat{r}_{o,k}(s) - r(s)| > \sqrt{\frac{\log(2SHk^2)}{2N_k(s)}} \right) \tag{P.29}$$

$$\leq H\sum_{k=1}^{K}\sum_{s \in \mathcal{S}}\frac{1}{SHk^2} \leq \frac{\pi^2}{6}, \tag{P.30}$$

where line (P.28) follows from the definition of the good event $G_k$. Line (P.29) is a union bound on the states. Line (P.30) comes from Höeffding's inequality.

For the first term (A) we define the event $E_k$ for all $k \in [K]$:

$$E_k = \left\{ \forall s \in \mathcal{S} : N_k(s) \geq (k-1)d(s) - \sqrt{\frac{k-1}{2}\log(SHk^2)} \right\}.$$

If this event holds then every state $s \in \mathcal{S}$ is visited at least $(k-1)d(s) - \sqrt{\frac{k}{2} \log(SHk^2)}$ times, where $d(s)$ is defined as in Lemma B.6.

Considering the term (A), we have:

$$\sum_{k=1}^{K} \mathbb{E}[\Delta_{I_k}|G_k] \leq \underbrace{\sum_{k=1}^{K} \mathbb{E}[\Delta_{I_k}|G_k, E_k]}_{(C)} + \underbrace{H \sum_{k=1}^{K} \Pr(\neg E_k)}_{(D)},$$

where we bound the in the second term $\Delta_{I_k} \leq H$.

We start bounding the second term (D). We apply Lemma B.6 after a union bound over the states:

$$H \sum_{k=1}^{K} \Pr(\neg E_k) = H \sum_{k=1}^{K} \Pr\left(\exists s \in \mathcal{S} : N_k(s) < (k-1)d(s) - \sqrt{\frac{k-1}{2} \log(SHk^2)}\right)$$

$$\leq H \sum_{s \in \mathcal{S}} \sum_{k=1}^{K} \Pr\left(N_k(s) < (k-1)d(s) - \sqrt{\frac{k-1}{2} \log(SHk^2)}\right)$$

$$\leq H \sum_{s \in \mathcal{S}} \sum_{k=1}^{K} \frac{1}{SHk^2} \leq \frac{\pi^2}{6}.$$

Now it remains to bound the term (C) that, using Lemma B.7, is zero whenever $\sqrt{\frac{\log(2SHk^2)}{2N_k(s)}} \leq \frac{\Delta_Q - c}{2SH}$. Thus, under the events $E_k$ and recalling that under Assumption 1 we have $d(s) \geq \epsilon$, we obtain:

$$\sqrt{\frac{\log(2SHk^2)}{2N_k(s)}} \leq \sqrt{\frac{\log(2SHk^2)}{2(k-1)\epsilon - \sqrt{2(k-1)\log(SHk^2)}}}.$$

From which, we derive the condition:

$$\overline{K} \geq 1 + \left(\frac{2H^2 S^2 \log(2SH\overline{K}^2)}{2(\Delta_Q - c)^2} + \sqrt{\frac{\overline{K}-1}{2} \log(SH\overline{K}^2)}\right) \frac{1}{\epsilon}.$$

Then, we take the infimum over $c$. Thus, for the term (C), we consider the decomposition:

$$\sum_{k=1}^{K} \mathbb{E}[\Delta_{I_k}|G_k, E_k] \leq \sum_{k=1}^{\overline{K}} \mathbb{E}[\Delta_{I_k}|G_k, E_k] + \sum_{k=\overline{K}+1}^{\infty} \mathbb{E}[\Delta_{I_k}|G_k, E_k] = \overline{K}\Delta + 0,$$

where we bounded $\Delta_{I_k} \leq \Delta$ with $\Delta = \max_{i \in [M]} \Delta_i$. Then the total regret is given by:

$$\mathbb{E}[\text{Regret}(K)] \leq \overline{K}\Delta + \frac{\pi^2}{3}.$$

$\square$

## C Adversarial agent

In this paragraph, we provide some hints about the adversarial case, to illustrate the additional complexities that arise. In the adversarial setting, the agent can play a different policy at each step, inside the set of possible policies that satisfy the SSE, namely $\Pi_i^{\text{SSE}}$. For the Af setting, to have bounded regret, we have to add the following assumption.

**Assumption 2.** *For all $i \in [M]$, let $\pi, \pi' \in \Pi_i^{\text{SSE}}$, then $d_h^{i,\pi}(s) > 0$ if and only if $d_h^{i,\pi'}(s) > 0$.*

Under this assumption (less strict than Assumption 1), Theorem 5.1 continues to hold. Indeed, though the agent can adversarially change the policies, we can still define the policy $\widehat{\pi}$, since the policies in the set $\Pi^{\text{SSE}}$ do not disconnect the reachable set of states. On the other hand, without this assumption, the algorithm needs some modifications, since the agent can stuck the configurator with actions that after some episodes will not be played any more; this behavior can lead to an estimated policy that visits unreachable states.

For the Rf, instead, under the following assumption (that is a natural extension of Assumption 1), Theorem 5.2 continues to apply.

**Assumption 3.** *There exists $\epsilon > 0$ such that:* $\min_{i \in [M]} \min_{s \in \mathcal{S}} \max_{h \in [H]} d_h^i(s) \geq \epsilon$, *where $d_h^i(s)$ is the probability of visiting the state $s \in \mathcal{S}$ at time $h \in [H]$ in configuration $p_i$ under every agent's best response policy $\pi_i \in \Pi_i^{SSE}$.*

In this case the reward continues to give the structure to connect the policies and the models. However, we believe that to solve the adversarial case without these assumptions would require modifying the algorithm, and it is left to future work.

# D   Experimental Details

In this appendix, we report additional experimental details and results.

## D.1   Configurable Gridworld

**Description**   In Figure 4 the environment of the Configurable Gridworld is shown. The configurable Gridworld is a $3 \times 3$ gridworld with an obstacle in the cell $(2, 2)$, which with a probability $p$ causes the agent action *right* not to be performed. The starting state is in every configuration $(1, 2)$ and the goal state is $(3, 2)$.

**Additional Experiments**   We report additional experiments for the Configurable Gridworld environment. For the Configurable Gridworld with size $3 \times 3$, horizon 10, we perform 4 experiments with an increasing number of configurations. In this case the expert policy is deterministic. Figure 5 shows the results of the experiments. We can notice that with more than 100 configurations AfOCL does not achieve constant regret in 5000 steps, instead RfOCL converges in every experiment.

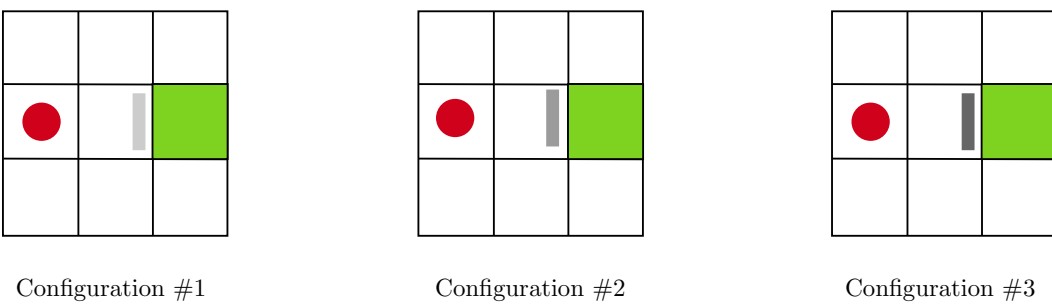

| Configuration #1 | Configuration #2 | Configuration #3 |

Figure 4: Configurable Gridworld: from left to right the 3 configurations represent increasing "power" of the obstacle.

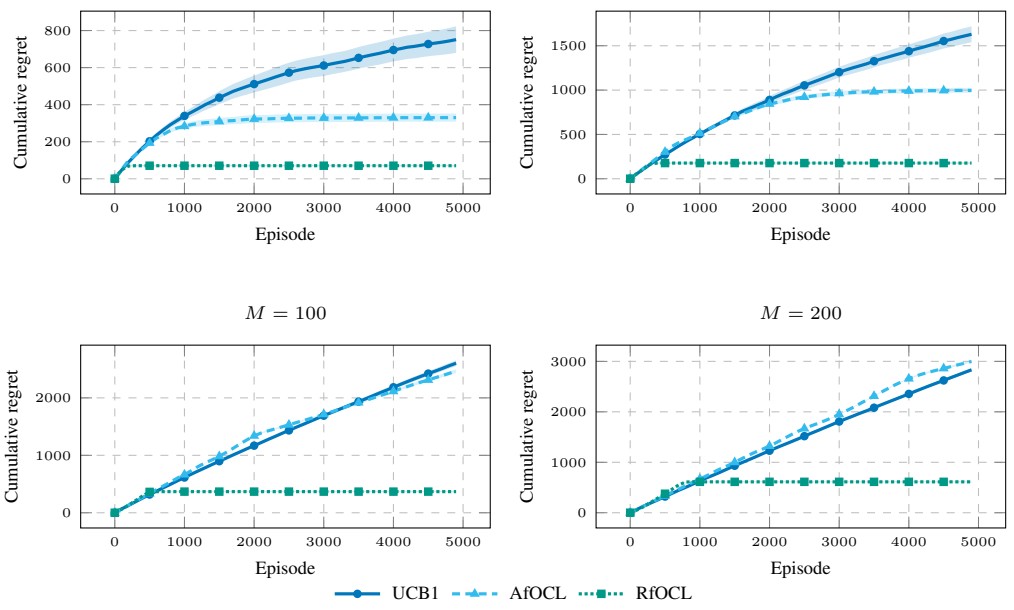

Figure 5: From left up to right down $10, 30, 100, 200$ configurations' number.

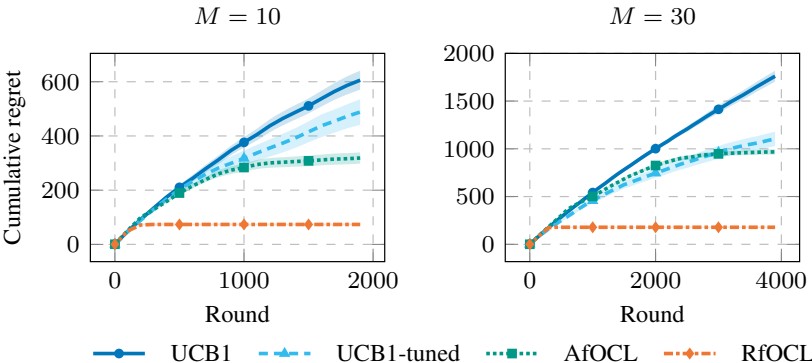

Figure 6: Cumulative regret for the Gridworld experiment. 50 runs, 98% c.i.

We report also the same experiment shown in the main paper with a tuned-version of UCB1 (see figure 6). We would like to underline that, also in this case, the two proposed algorithm AfOCL and RfOCL achieved a constant behavior while UCB1-tuned has a logarithmic behavior.

### D.2 Market

A Configurable Market is a simplified model for a marketplace. The agent, namely the customer, wants to buy a given set of products $Q_A$ in the minimum number of steps. Instead, the configurator has the role in placing all the products $Q \supset Q_A$ in the marketplace to maximize the market's revenue inducing the agent to buy other products in addition to those it would buy. The configurator's reward is $1$ any time the agent passes over a state where a product is placed and $0$ in all the other states. Whereas the agent's reward is $-1$ everywhere and gains a bonus of $0.9$ when it passes over a state with a product in $Q_A$. In other words, the products remain fixed in the market, and the configurator can change the transition model within a set of random transition models. However, from an abstract point of view, this is equivalent to moving the products in the Gridworld.

In Figure 7 the market domain with 3 different configurations is shown. The market domains consists in $K \times K$ states, where every product is assigned to a specific state. The configurator can change the transition matrix for all the states except for the starting state and the "exit" state. Every different configuration can be thought as shuffling the cells of a gridworld.

In Figure 8, AfOCL and RfOCL are compared against UCB1. The number of configurations is 10, the horizon 15, and the Gridworld size is $4 \times 4$. In every run, we construct 10 different transition models, which specify the 10 configurations. Also, in this experiment, the trend is confirmed since AfOCL and RfOCL outperform UCB1. We observe that the two algorithms, in this environment, behave similarly, and this is due to the small number of configurations. However, we can notice RfOCL at the end of the considered episodes approaches the constant regret.

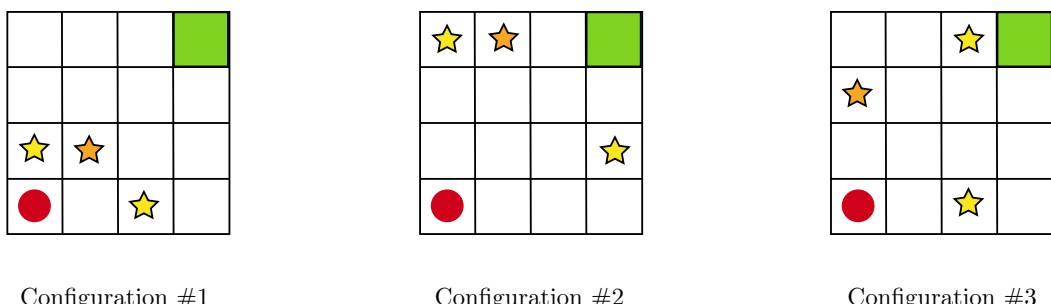

Configuration #1          Configuration #2          Configuration #3

Figure 7: Market: the figure shows a $5 \times 5$ market. The red state is the starting state, instead the green state is the "end" state. The stars are the product and the orange star is the only product the agent is interested in.

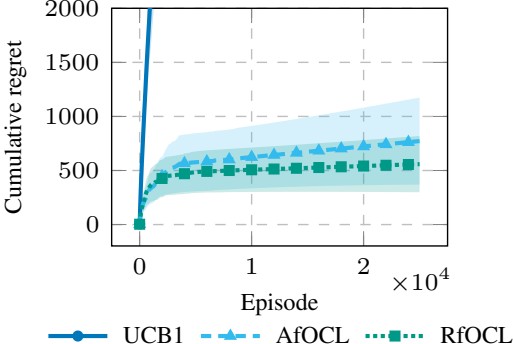

Figure 8: Cumulative regret as a function of the episodes for the Configurable Market experiment. 50 runs, 98% c.i.

### D.3 Teacher Student

In Figure 9 an illustrative example of the Teacher-Student domain is reported. Right arrows correspond to answer No, and green arrows to answer Yes. The transparency is due to the level of probability of every transition. The configurator can change the transition matrix for the answer Yes, instead the transition matrix for action No is fixed for all the configurations.

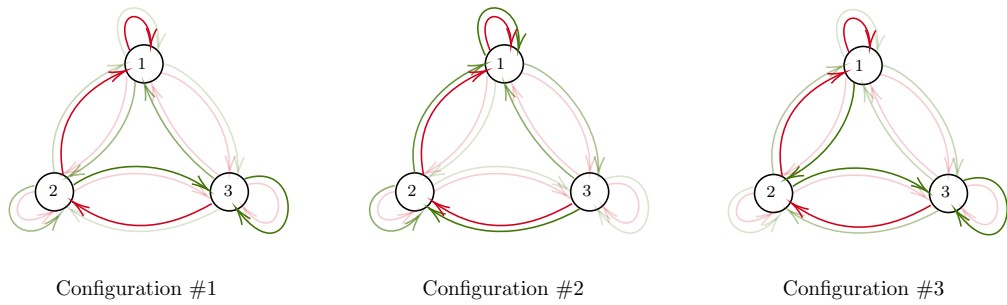

Configuration #1          Configuration #2          Configuration #3

Figure 9: Teacher Student.