# OpenReview forum: "Learning in Non-Cooperative Configurable Markov Decision Processes"
_NeurIPS.cc/2021/Conference — NeurIPS 2021 Poster_

### Official Review · Reviewer_Qthd · 2021-07-12

**Rating:** 7
**Confidence:** 3

**Summary:**

This paper proposes an extension to the configurable MDP framework, which implicitly assumes a shared reward function by the acting agent and the configurator, to the non-cooperative case where the reward functions of the two agents may be different and unknown to the other. Modeling the problem as a leader follower game, the papers present two online regret-minimization algorithms based on a multi-armed bandit strategy to learn the best environment configuration under different conditions. In the first, the configurator has access only to observed state-action traces (AfOCL), and in the second the configurator additionally observes noisy reward signals (RfOCL). The core idea of the algorithms is to iteratively prune the space of "plausible policies" of the acting agent based on observed feedback to compute an increasingly accurate optimistic value function for each environmental configuration for the configurator at the start of each episode. The authors prove that both algorithms incur finite expected regret, and show that regret scales linearly in the number of configurations for AfOCL, and is independent of the number of configurations in the case of RfOCL. Finally, the authors provide empirical results of the performance of AfOCL and RfOCL against the baseline UCB1 in two experimental domains, a gridworld domain and a teacher-student domain.

**Limitations And Societal Impact:**

This work assumes that the act of reconfiguring the environment does not have a cost to either configurator or acting agent. As the motivating domains are real-world problems with humans, if the environment is reconfigured dozens or hundreds of time for a single acting agent, possibly diminishing the cumulative reward earned by the agent in some configurations, this assumption may be violated. This assumption is also made in regular Conf-MDPs, but is perhaps more critical in the non-cooperative case where the reconfigurations are not always benefiting the human. Perhaps the authors could touch on this more.

**Main Review:**

Primary Contributions:

(1) Formal extension of the Conf-MDP model to the non-cooperative case (NConf-MDP).

(2) Formalization of the problem as a leader follower game under Stackelberg equilibrium criteria.

(3) Two online regret-minimization algorithms, AfOCL and RfOCL.

(4) Proofs of bounded expected regret for both algorithms, the first linear in the number of configurations, the second independent of that.

(5) Experimental evaluations on two domains, a gridworld domain and a student-teacher domain, against the baseline UCB1.

Pros:

(1) Configurable MDPs are a recent addition to the community and well-motivated by many real-world problems. The non-cooperative setting considered in this paper is the natural extension of this framework and of interest to the community.

(2) The overall structure was clear and the main ideas were well-explained.

(3) The algorithms presented in the paper are simple and easily understood, but are intelligent in their approach and use of existing insights in related areas of multi-armed bandits and stackelberg games.

(4) The authors provide rigorous theoretical guarantees on their algorithms, and the simulated results appear to match these results for the most part.

(5) The algorithms perform well and significantly outperform the baseline, UCB1, in the experimental domains considered.

Cons:

(1) Discussion of certain areas in the related work is lacking. In particular, the paper would benefit from a more detailed discussion of [45] and the broader area of environment design/shaping as it is central to the problem considered in the paper. We suggest some additional related work:

 --- "Equi-Reward Utility Maximizing Design in Stochastic Environments." Keren et al. IJCAI 2017

 --- "Mitigating Negative Side Effects via Environment Shaping." Saisubramanian and Zilberstein. AAMAS 2021.

(2) Page 5, paragraph on "Regret Guarantees". The authors claim that "either an (s, h) pair is visited with high probability, or it will impact only marginally on the performance". Something is unclear, or perhaps incorrect, about this statement. A deterministic policy in a stochastic domain could to impactful states being visited infrequently. For instance, a state (say at timestep 1), with a small probability of being reached but with an arbitrarily high negative reward (e.g. a dead-end). Perhaps there is room for clarification.

(3) Experiments are performed on very simple and small domains that are not entirely convincing.

Additional Comments:

- In figure 1, should "round" be "episode"?

- Is there any intuition to the term \overline{K} in Theorem 5.2? As presented, it is hard to interpret the RfOCL regret in a meaningful way.

**Time Spent Reviewing:**

6

---

> ### Author Response · Authors · 2021-08-09
> **Answer to Reviewer Qthd**
>
> We thank the reviewer for the insightful comments. We address the “Cons” in the following:
>
> (1) Thank you for reporting the interesting related works; we will surely include them in the “Related Works” section and extend the current discussion. We agree with the reviewer about the connection between Configurable MDPs and environment design/shaping. At a high level, the two frameworks share analogous objectives: they both aim at determining an environment with a certain goal that can differ from that of the agent. We take the liberty to stress that there are some notable differences. In particular, the classical environment design formulation [1] assumes that the configurator (called “interested party”) knows the agent’s best response function, while in our approach, we learn it by interaction. Nevertheless, the general environment design makes no assumption about the underlying environment (it is not required to be an MDP). Instead, [2] limits to MDPs and considers a form of cooperative environment design in which the goal is to maximize the agent’s performance. Interestingly, [2] and [3] also account for a cost function to penalize expensive environment configurations.
>
> (2) The meaning of the sentence is that, for analysis purposes, we can partition the set of states into two subsets based on the probability of being visited under the agent’s best response policy, having defined a suitable probability threshold (Lemma B.2). The states that are visited with a probability below the threshold will have a negligible impact on the regret and, thus, are discarded. This is possible because the reward function is bounded in $[0,1]$ (so it is not possible to have arbitrarily negative rewards). The reward boundedness is a widespread assumption in the RL literature when performing theoretical analysis of algorithms [e.g., 5]. We will clarify this point in the paper.
>
> (3) We are aware that the experiments are performed in simple domains. Nonetheless, the goal of the simulations is to provide empirical validation of the proposed algorithms by highlighting their properties in terms of constant cumulative regret, as opposed to, for instance, UCB1. Being this the first work about Non-Cooperative Con-MDPs, we think that practical versions of the algorithms, able to scale on more complex settings, should be regarded as future works.
>
> **Additional Comments**
> * A round corresponds to an episode; we will replace the label with “episode” for clarity.
> * Given that the form of the inequality is $\overline{K} \ge O(c  \sqrt{\overline{K} \log \overline{K}})$, using standard results (like Lemma 12 of [4]), we can deduce that $\overline{K} \le O(c^2 \log c)$. We will discuss this point in the paper.
>
> **Limitations and Societal Impact**
> Thank you for pointing out an interesting argument. Indeed, we are aware that reconfiguring the environment is an activity that typically leads to higher costs compared with policy learning. We did not consider this aspect in the formalization of the Non-Cooperative Conf-MDP since it would possibly make the problem more complex (like when considering bandits with switching costs). Nevertheless, it represents an interesting direction and surely a point of discussion about societal impact.
>
>
> [1] Zhang, Haoqi, Yiling Chen, and David C. Parkes. "A general approach to environment design with one agent." In Twenty-First International Joint Conference on Artificial Intelligence. 2009.
>
> [2] Keren, Sarah, Luis Enrique Pineda, Avigdor Gal, Erez Karpas, and Shlomo Zilberstein. "Equi-Reward Utility Maximizing Design in Stochastic Environments." In IJCAI. 2017.
>
> [3] Saisubramanian, Sandhya, and Shlomo Zilberstein. "Mitigating Negative Side Effects via Environment Shaping." In Proceedings of the 20th International Conference on Autonomous Agents and MultiAgent Systems, pp. 1640-1642. 2021.
>
> [4] Jonsson, Anders, Emilie Kaufmann, Pierre Menard, Omar Darwiche Domingues, Edouard Leurent, and Michal Valko. "Planning in Markov Decision Processes with Gap-Dependent Sample Complexity." Advances in Neural Information Processing Systems 33 (2020).
>
> [5] Jaksch, Thomas, Ronald Ortner, and Peter Auer. "Near-optimal Regret Bounds for Reinforcement Learning." Journal of Machine Learning Research 11, no. 4 (2010).

---

> > ### Comment · Reviewer_Qthd · 2021-08-25
> > **Response Paper 6820**
> >
> > The answers provided by the authors were helpful and satisfactory, and did not substantively alter my overall rating of the paper. I believe that including these clarifications and explanations in their paper when possible would serve to strengthen the overall paper.

---

### Official Review · Reviewer_zQVR · 2021-07-15

**Rating:** 6
**Confidence:** 3

**Summary:**

The paper studies learning in configurable MDPs with two-agents, a reinforcement learning agent and a configurator that can modify the environment parameters. In contrast to prior work, the paper considers a setting in which the configurator and the agent can have misaligned objectives. The setting is formally modeled as a Stackelberg game, in which the configurator is the leader that  selects a transition model and the agent is the follower that plays the best response policy under this model. The goal is to design a learning algorithm for the leader with provable regret guarantees. The paper provides two learning algorithms for the configurator and formally analyzes their regret properties, which turn out to be better than those of a naive solution based on UCB1. The paper supports its theoretical claims using simulation-based experiments.

**Limitations And Societal Impact:**

The checklist states that the limitations are addressed in Section 8 (conclusion), but the conclusion only summarizes the paper and provides one sentence description of a future work. Since this is a theory paper, a discussion about limitations could explain the limitations of the modeling assumptions. Potential negative societal impacts are also not discussed, but given the topic of the paper, I don't find this necessary.


**Main Review:**

Overall, I enjoyed reading this paper and I find its results interesting and relevant. On the other hand, some aspects of the paper could be improved, e.g., in terms of clarity, and additional justification for the modeling assumptions could be provided. More detailed comments are below.

Novelty: The paper studies a novel problem setting, building on the work of [29,27]. The setting has a clear motivation and the paper provides a good overview of the related work.  That said, the setting seems to be similar to adversarial attacks. How does this work relate to adversarial attacks in RL (e.g., adversarial attacks on policies 'Huang et al. Adversarial attacks on neural network policies', policy poisoning attacks 'Ma et al., Policy poisoning in batch reinforcement learning and control', attacks on environment parameters (rewards/transitions) 'Rakhsha et al., Policy Teaching via Environment Poisoning: Training-time Adversarial Attacks against Reinforcement Learning', etc.)?

Quality: In my opinion, the results of this paper advance the work on configurable environments, and the reasoning behind them is clearly explained. However, there are a few points that could be discussed in more detail:
- The setting of the paper restricts the leader (configurator) to choose from a finite set of transition models. I think the implications of this restriction are not clearly stated in the text. For example, would the hardness result of Letchford et al. 'Computing Optimal Strategies to Commit to in Stochastic Games' apply in a more general case that doesn't place this restriction?
- In general, the setting is closely related to stochastic games. This connection is mentioned and partly discussed in Section 6, but it could be extended. For example, can we model this setting as a Stackelberg stochastic game (e.g., where the leader has an influence on transitions via its policy)? The paper highlights that it is the first to provide problem-dependent regret bounds in multi-agent RL. On the flip side, the setting of this paper seems more restrictive than stochastic games, which is a standard framework for multi-agent RL.
- Some of the variable/factors in the regret bounds could be more thoroughly explained. For example, it seems that the values of $\rho$ in Eq. (3) or $\hat K$ can be quite large, which could make the corresponding factors in the regret bounds bounds 'impractical'. Some discussion on this would be useful.

Clarity: The structure of the paper is good, but some parts could be improved in terms of clarity as they contain typos or are ambiguous. Some notable examples include:
- The equation in line 158, should $\mathcal A_{i}^{k,h}(s)$ be a set of actions containing $\pi(s)$?
- The sentence in lines 188-187, why is the regret bound in (3) independent of \Delta_i?
- The sentence in lines 283-284, how does offline setting relate to controlling all of the agents?

Significance: The paper provides two new learning algorithms for a configurable MDP setting and derives non-trivial regret bounds for them. These results are potentially relevant for other sub-areas of RL, including environment design and multi-agent RL.


**Time Spent Reviewing:**

4/5 hours

---

> ### Author Response · Authors · 2021-08-09
> **Answer to Reviewer zQVR**
>
> We would like the reviewer for the useful suggestions. We answer the raised points in detail.
>
> **Novelty**
> We thank the reviewer for mentioning the interesting related works; we will surely include them in the “Related Works” section and extend the current discussion. The Non-Conf setting is related to adversarial attacks since in this setting, the rewards and transitions are manipulated by an external entity. However, there are substantial differences between the two frameworks. The interaction between the agent and the configurator happens in a hierarchical way and the configurator can only change the transition model at the beginning of an episode between a finite set of possible configurations. Moreover, in our setting, we do not consider “adversarial” intentions between the configurator and the agent; these two entities play to optimize their own reward function. Finally, in *policy poisoning* (a form of adversarial attack) the adversary induces the agent to learn a given target policy, while in our setting, the configurator acts with the goal of maximizing its own reward function.
>
> **Quality**
> - We think that the hardness results of Letchford et al. [1] do not apply to our setting, since the setting considered in [1] regards infinite-horizon stochastic games problems, and not finite-horizon ones, as we consider in our work. However, we agree that one limitation of our work is considering only a finite set of configurations. We think that considering a compact set of configurations can be an interesting future direction that needs more investigation.
> - Stochastic games are related to our setting, although we cannot model the problem using a stochastic game. In fact, agents and configurators act at a different level, and an agent cannot directly change the transition probability function of the MDP (directly changing the transition model is inherently more powerful than influencing the transition probabilities using a policy). However, we think that the technical ideas that we used in the Non-Conf setting can be of inspiration also for stochastic general-sum games. We will replace in the introduction the word “Multi-Agent RL” with “multi-entity RL”, as we have written in the conclusions, and we will extend the discussion on the relation between Non-conf MDPs and Stochastic Games.
> - The term $\rho$ can be large if the policy is stochastic and we are approximating it with a deterministic policy prescribing an action played in the original policy with a small probability. For technical reasons, we need to use as deterministic policy the one constructed for every pair (s,h) recording the first action seen in (s,h). Instead, if the agent’s policy is deterministic $\rho$ is always equal to 1.
> The term $\overline{K}$, instead, depends on the minimum suboptimality action-gap of the agent, then it can be arbitrarily large (like in the problem-dependent regret bound of UCB1). The analysis focuses on the relation between the two entities and how they contribute to regret. However, we would like to emphasize that the regret is also bounded by the one of UCB1. See also answer to Reviewer Qthd for more insights on how to obtain an explicit form of $\overline{K}$.
>
> **Clarity**
> - If the agent acts deterministically $\mathcal{A}^i_{k,h}(s)$ is a singleton if s has been already visited at iteration $k$ and step $h$. For this reason we wrote that $\mathcal{A}^i_{k,h}(s) = \pi(s)$, although it is more correct to write $\mathcal{A}^i_{k,h}(s) = \\{ \pi(s) \\}$. We corrected it.
> - The sentence argues that the part concerning the AfOCL regret $MH^3S^2\rho$ is independent of $\Delta_i$. We clarified the sentence.
> - In the works [2,3,4,5] the authors consider the so-called *offline* setting [5,6]. In this case, the authors assume to act in a self-play setting, where all the agents play the same algorithm.
>
>
> **Limitations and societal impact**
> We will add in the final version of the paper a discussion on the limitations of the proposed algorithms.
>
> [1] Letchford, J., MacDermed, L., Conitzer, V., Parr, R., & Isbell, C. L. (2012, July). Computing optimal strategies to commit to in stochastic games. In Twenty-Sixth AAAI Conference on Artificial Intelligence.
>
> [2] Liu, Q., Yu, T., Bai, Y., & Jin, C. (2021, July). A sharp analysis of model-based reinforcement learning with self-play. In International Conference on Machine Learning (pp. 7001-7010). PMLR.
>
> [3] Bai, Y., & Jin, C. (2020, November). Provable self-play algorithms for competitive reinforcement learning. In International Conference on Machine Learning (pp. 551-560). PMLR.
>
> [4] Zhang, K., Kakade, S. M., Başar, T., & Yang, L. F. (2020). Model-based multi-agent rl in zero-sum markov games with near-optimal sample complexity. arXiv preprint arXiv:2007.07461.
>
> [5] Xie, Q., Chen, Y., Wang, Z., & Yang, Z. (2020, July). Learning zero-sum simultaneous-move markov games using function approximation and correlated equilibrium. In Conference on Learning Theory (pp. 3674-3682). PMLR.
>
> [6] Wei, C. Y., Hong, Y. T., & Lu, C. J. (2017). Online reinforcement learning in stochastic games. arXiv preprint arXiv:1712.00579.

---

### Official Review · Reviewer_bBaH · 2021-07-16

**Rating:** 6
**Confidence:** 3

**Summary:**

The authors provide an algorithm for online learning in a non-cooperative configurable MDP. They propose a setting where a configurator can choose a transition model for the MDP from a finite set of models. The goal is then to observe the agent's behaviour (either the state-action sequences or a noisy reward) and to learn an optimal configuration that results in maximum configurator reward given that the agent behaves optimally with respect to its own reward function.

**Limitations And Societal Impact:**

This work is mainly theoretical in nature and is unlikely to have an immediate social impact. However, the authors do provide some examples (such as a supermarket owner optimizing their shelves for profit) in which their work may be applied. It would be worth a small discussion to address if these applications for the work in this paper is actually a negative impact. For example, following on with the example provided in the introduction, is it a social good that we provide more methods for businesses to manipulate user behaviour to optimize their profits? Can this work be used in social network applications to increase user engagement by showing extremist or misleading content even if it is not the users' original intent?

**Main Review:**

The paper is written clearly. However, some of the notation and concepts can be a bit dense and a use of a toy example or two can greatly help readability of the paper. Additionally, I feel the problem is not sufficiently motivated. The examples provided in the introduction are all well studied applications with numerous approaches. The authors do not do enough to explain why their new approach is necessary compared to past work.

As noted by the authors, the work in this paper is really similar to the multi-armed bandit formulation. However, the authors argue that in most bandit formulations, the regret is unbounded as they do not incorporate the agent's observed policy or reward into the algorithm as is done in this paper. While this is true, I don't know if the novelty in this paper is sufficient for a publication. As far as I can tell the approach is a relatively straightforward extension of using the observed actions or rewards of the agent to estimate the value function and choosing the corresponding best configuration.

Overall, if the authors clarify the above point, I think the paper is interesting, and I believe has an incremental contribution to the literature. However, I believe it is not sufficiently well motivated and the contribution is not yet strong enough for publication.

Response: I appreciate the authors' response and they addressed my issues with the multi-armed bandit and novelty.




**Time Spent Reviewing:**

6

---

> ### Author Response · Authors · 2021-08-09
> **Answer to Reviewer bBaH**
>
> We would like to thank the reviewer for the insightful suggestions. We answer the raised points below.
>
> **Motivations**
> We believe that the formalism of the Non-Cooperative Conf-MDPs captures a form of interaction that cannot be effectively modeled with existing approaches and in particular, neither with (cooperative) Conf-MDPs and Stochastic Games. First, while (cooperative) Conf-MDPs account for the presence of a configurator in charge of configuring the environment and an agent, these two entities share the same goal, modeled by a unique reward function. Instead, we are interested in modeling the possibility of a configurator with an objective conflicting with that of the agent. Second, Stochastic Games (SG) allow the modelization of multiple agents interacting in an environment, with possibly conflicting objectives. However, each agent has its own policy and the environment is normally fixed. Differently, in our setting, agent and configurator act at different levels. Indeed, the configurator can modify the transition probabilities of the environment (so the configuration activity is inherently more powerful compared with a policy). Finally, we see the configurator-agent interaction in a hierarchical way, in which the agent might even be unaware of the configurator presence, making the SG formulation quite inappropriate. We will clarify these points in the paper.
>
> **Relations with multiarmed bandit and novelty**
> As noted by the reviewer, the problem can be formulated as a multiarmed bandit problem. The formulation of the algorithms follows principles already used in literature [1,2,3]. However, the algorithms contain novel ideas to exploit the *structure* of the problem by incorporating information from both entities: the configurator and the agent. In fact, as far as we know, this is the first paper that uses the sub-optimality gaps of an uncontrollable entity (the agent) with the ones of the controllable entity (the configurator). Indeed, by considering only the configurator (and disregarding the agent presence) in the analysis we cannot derive the constant regret bound. Finally, as far as we know, we provide the first problem-dependent regret upper bound in a multi-entity setting. These novel ideas could be of inspiration also for the multi-agent community.
>
> **Limitations and Societal Impact**
> Thank you for pointing out an important aspect that we have not mentioned in the paper.
> Undoubtedly, methods that incentivize the manipulation of users' behavior can have a negative societal impact. We will add a section in the paper discussing the potential negative impact that methods, that can be used for a marketing campaign, can have on society.  However, we would like to underline that, as the reviewer noticed, the work is mainly theoretical and, at the present level, can hardly be used in a malevolent way.
>
> [1] Jaksch, T., Ortner, R., & Auer, P. (2010). Near-optimal Regret Bounds for Reinforcement Learning. Journal of Machine Learning Research, 11(4).
>
> [2] Jin, C., Allen-Zhu, Z., Bubeck, S., & Jordan, M. I. (2018, December). Is Q-learning provably efficient?. In Proceedings of the 32nd International Conference on Neural Information Processing Systems (pp. 4868-4878).
>
> [3] Azar, M. G., Osband, I., & Munos, R. (2017, July). Minimax regret bounds for reinforcement learning. In International Conference on Machine Learning (pp. 263-272). PMLR.

---

### Official Review · Reviewer_LFno · 2021-07-17

**Rating:** 7
**Confidence:** 1

**Summary:**

This paper presents an extension of the existing Configurable-MDP framework to a non-cooperative setting; one where the agent and the configurator could also have different reward functions. This setting (referred to as Non-cooperative Configurable MDPs) allows for modelling a wider range of situations, which also include the scenario where the agent and configurator display non-cooperative behaviour. The problem-setting is modelled as a leader-follower game, with the configurator as the leader and the agent as the follower. The authors present two algorithms for the configurator: AfOCL- which assumes that during online interaction the configurator can observe the agent's actions; RfOCL- which assumes that during online interaction the configurator can observe the agent's actions and obtains noisy estimates of its reward as well. The authors go on to prove that AfOCL achieves finite expected regret, which scales *linearly* with the number of possible configurations. This is already an improvement over the standard UCB algorithm which yields a regret that grows indefinitely over time. They also show that RfOCL the regret does not depend on the number of configurations under the assumption that the agent has a non-zero probability in some time-step of the horizon to visit every state in the MDP. The experimental evaluations were carried out on two environments: A configurable gridworld, and a simulated teacher-student environment. In both settings, their algorithms obtain constant  cumulative regret, as opposed to the logarithmic regret suffered by the UCB algorithm, validating their result.

**Limitations And Societal Impact:**

The paper explicitly states its limitations. As the paper mainly has theoretical results, it potentially has no negative societal impact.

**Main Review:**

**Pros**

* The paper was very clearly written and all the terminology well explained before they were presented.
* The experimental setup was very clean, and the results presented provided insights into the workings of the 2 proposed algorithms. The fact that both stochastic and deterministic policies of the agent were considered, validates the performance of the algorithms in all settings.
* The claim in the paper is that the regret bounds presented are the first problem-dependent regret results for multi-entity MDPs. This result can be exploited in the multi-agent adversarial RL setting to design more robust policies.

**Cons**

* Experimental Evaluations: The evaluation on the 2 environments presents proof of concept necessary to validate the theoretical results. However, it would be interesting to empirically see the performance of the algorithms on a purely adversarial setting, and get some further insights into worst-case performance in such a setting. Maybe illustration of the algorithms on such a setting could be an interesting add-on.

**General Comments** Overall, the paper was well structured and the ideas were coherent.

*Originality*: Novel

*Clarity*: Clear

*Quality*: Good

*Significance*: High


**Time Spent Reviewing:**

3

---

> ### Author Response · Authors · 2021-08-09
> **Answer to Reviewer LFno**
>
> We would like to thank the reviewer for the supportive review.
>
> We take the opportunity to provide clarification about the *adversarial setting*. By *adversarial setting*, we indicate the possibility that the agent selects a policy adversary but within the set of policies satisfying the strong Stackelberg equilibrium. Therefore, in this setting, the agent has limited freedom in the policy selection. Instead, the *purely adversarial* setting, in which the agent can even decide to play a policy that *does not* induce the strong Stackelberg equilibrium (for instance, to threaten the configurator), needs a different formulation of the regret objective and, we believe, is out of the scope of this work. We thank the reviewer to point out this interesting direction that may be investigated in the future.

---

### Decision · Program_Chairs · 2021-09-27

**Decision:**

Accept (Poster)

**Comment:**

Overall, the reviewers are fairly positive on this paper, which offers a novel take on the “environment design” problem in the non-cooperative (but not full adversarial) Stackelberg-type setting, using the framework of configurable MDPs. This new formulation is rather interesting, the (problem-dependent) regret analysis and the empirical confirmation offered are also valuable (though the empirical setup is critiqued as rather simple). The author response alleviated some of the reviewers questions and concerns, especially with regard to the problem formulation and its motivation There reviewers make some useful suggestions for improving the clarity of the paper, and related work suggestions. These should be addressed by the authors in revision.